# Flat-LoRA: Low-Rank Adaptation over a Flat Loss Landscape

**Tao Li** [* 1]  **Zhengbao He** [* 1]  **Yujun Li** [2]  **Yasheng Wang** [2]  **Lifeng Shang** [2]  **Xiaolin Huang** [1]

## Abstract

Fine-tuning large-scale pre-trained models is prohibitively expensive in terms of computation and memory costs. Low-Rank Adaptation (LoRA), a popular Parameter-Efficient Fine-Tuning (PEFT) method, offers an efficient solution by optimizing only low-rank matrices. Despite recent progress in improving LoRA's performance, the relationship between the LoRA optimization space and the full parameter space is often overlooked. A solution that appears flat in the loss landscape of the LoRA space may still exhibit sharp directions in the full parameter space, potentially compromising generalization. We introduce Flat-LoRA, which aims to identify a low-rank adaptation situated in a flat region of the full parameter space. Instead of adopting the well-established sharpness-aware minimization approach, which incurs significant computation and memory overheads, we employ a Bayesian expectation loss objective to preserve training efficiency. Further, we design a refined strategy for generating random perturbations to enhance performance and carefully manage memory overhead using random seeds. Experiments across diverse tasks—including mathematical reasoning, coding abilities, dialogue generation, instruction following, and text-to-image generation—demonstrate that Flat-LoRA improves both in-domain and out-of-domain generalization. Code is available at https://github.com/nblt/Flat-LoRA.

## 1. Introduction

Pre-training followed by fine-tuning has become the dominant paradigm in modern machine learning, achieving state-

---

[*]Equal contribution. This work was conducted when Tao was an intern at Huawei Noah's Ark Lab. [1]Department of Automation, Shanghai Jiao Tong University, Shanghai, China [2]Huawei Noah's Ark Lab. Correspondence to: Xiaolin Huang <xiaolinhuang@sjtu.edu.cn>.

*Proceedings of the 42$^{nd}$ International Conference on Machine Learning*, Vancouver, Canada. PMLR 267, 2025. Copyright 2025 by the author(s).

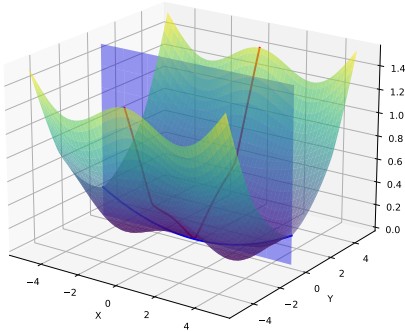

Figure 1: Illustration of LoRA optimization space. LoRA constrains optimization to a lower-dimensional space (**blue**). A flat minimum in LoRA space (**blue** curve) may exhibit sharp directions in the full parameter space (**red** curve).

of-the-art performance by leveraging the versatile capabilities of pre-trained models (Girshick et al., 2014; Kolesnikov et al., 2020; Radford et al., 2021; Li et al., 2024c). However, the enormous size of these models makes fine-tuning all parameters resource-intensive. Recently, Low-Rank Adaptation (LoRA) (Hu et al., 2022) has been proposed to address this challenge. LoRA fine-tunes only low-rank matrices, which can be merged with the pre-trained weights after training, incurring no extra overhead during inference. This approach significantly reduces trainable parameters, thereby lowering both training and storage requirements.

Many methods have been proposed to enhance LoRA performance, such as adaptive rank allocation (Zhang et al., 2023a), decomposition of optimization into direction and magnitude (Liu et al., 2024), and improved initialization strategies (Meng et al., 2024; Wang et al., 2024). Despite the promising potential these methods offer, the connection between the LoRA optimization space and the original full parameter space is often overlooked. Essentially, LoRA constrains optimization to a much lower-dimensional space, and its performance depends on how solutions in this restricted space relate to the full parameter space since the merged weights are ultimately used during inference. As illustrated in Figure 1, a flat minimum in the LoRA space may contain sharp directions in the view of the full parameter space, potentially leading to performance degradation. Figure 6 further demonstrates this phenomenon, revealing

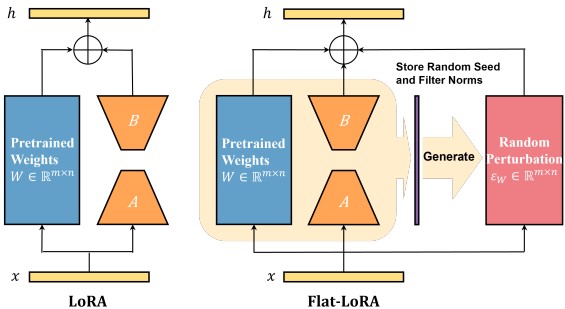

Figure 2: Illustration of LoRA (**Left**) and Flat-LoRA (**Right**). By introducing designed random weight perturbations during fine-tuning, Flat-LoRA identifies a low-rank solution that is flat in the loss landscape of the full parameter space. Unlike SAM, it eliminates the need for additional gradient steps and remains memory-efficient by storing only the random seed and a small number of filter norms (less than $1/r$ of the LoRA parameters for rank $r$).

the sharpness of loss landscape for the minima found by LoRA when examined in the full parameter space.

Flat minima in the loss landscape are widely believed to improve generalization and increase robustness to distribution shifts between training and test data (Hochreiter & Schmidhuber, 1994; 1997). This understanding has inspired Sharpness-Aware Minimization (SAM) (Foret et al., 2020), which is formulated as a min-max problem and has achieved state-of-the-art generalization. While integrating SAM with LoRA (referred to as LoRA-SAM (Li et al., 2024a)) for large model fine-tuning is promising, there are several issues that should be discussed. First, LoRA-SAM can only optimize the sharpness of the loss landscape in a restricted space (Section 3.2), which may not effectively improve generalization. Second, SAM requires an additional gradient step, doubling the training cost and rendering it impractical for large models. Lastly, computing sharpness in the full parameter space necessitates calculating gradients and storing perturbations for all weights, which contradicts the principles of parameter-efficient fine-tuning.

To address these challenges, we propose employing the Bayesian expectation loss objective (Duchi et al., 2012; Bisla et al., 2022) to smooth the loss landscape, thereby achieving flat minima in the full parameter space. Our approach, termed **Flat-LoRA**, leverages efficient random weight perturbations that can be stored as random seeds. In contrast to SAM, which requires additional gradient steps and maintaining an extra copy of model weights, Flat-LoRA ensures both time and memory efficiency. Moreover, we introduce refined perturbation generation strategies that consider weight magnitude and model width scaling, resulting in improved generalization performance.

Our main contributions can be summarized as follows:

- We find that low-rank adaptation may exhibit sharper loss landscapes in the full parameter space, prompting us to propose Flat-LoRA to mitigate this sharpness.

- We employ Bayesian expected loss with designed random weight perturbations to pursue flat minima, seamlessly integrating with existing methods while maintaining computational and memory efficiency.

- Extensive experiments across various natural language processing and computer vision tasks demonstrate that Flat-LoRA significantly improves both in-domain and out-of-domain generalization.

## 2. Related Work

### 2.1. Flat Minima and Generalization

The connection between the flatness of local minima and generalization has received much attention (Hochreiter & Schmidhuber, 1997; Chaudhari et al., 2017; Keskar et al., 2017; Dinh et al., 2017; Izmailov et al., 2018; Li et al., 2018b; Wu et al., 2020; Kwon et al., 2021; Zhuang et al., 2022; Li et al., 2024e). Recently, many works have tried to improve generalization by seeking flat minima (Tsuzuku et al., 2020; Zheng et al., 2021; Bisla et al., 2022). For example, Chaudhari et al. (2017) propose Entropy-SGD to search for flat regions by minimizing local entropy. Wen et al. (2018) design SmoothOut framework to smooth out the sharp minima. Notably, Sharpness-Aware Minimization (SAM) (Foret et al., 2020) establishes a generic training scheme for seeking flat minima by formulating a min-max problem and encouraging parameters sitting in neighborhoods with uniformly low loss, achieving state-of-the-art generalization improvements across various tasks. However, SAM doubles the training time compared to regular training, limiting its applicability to large-scale training.

Another branch of methods for recovering flat minima involves minimizing the expected Bayesian training loss under random weight perturbation (RWP), which is efficient and doesn't require additional gradient step (Bisla et al., 2022). Wang & Mao (2021) propose Gaussian model perturbation as a regularization scheme for improving SGD training, but it remains inefficient for multiple noise sampling. Bisla et al. (2022) connect the smoothness of loss objective to generalization and adopt filter-wise random Gaussian perturbation generation to recover flat minima and improve generalization. Li et al. (2022c; 2024d) further enhance the generalization performance of RWP by introducing an adaptive perturbation generation strategy and a mixed loss objective. Wu et al. (2022); Li et al. (2024b) demonstrate that injecting small random noise before or during fine-tuning can improve generalization. However, when applying to parameter-efficient fine-tuning, we must be mindful of the additional memory costs they may introduce.

## 2.2. Low-rank Adaptation and Variants

Recent studies have shown that the intrinsic dimension required for optimizing deep neural networks (DNNs) can be significantly lower than the total number of parameters (Li et al., 2018a; Gur-Ari et al., 2018). Notably, Li et al. (2022a) demonstrate the low-dimensional properties of DNNs' training dynamics, which have been leveraged to mitigate overfitting issues in adversarial training (Li et al., 2022b). Low-Rank Adaptation (LoRA) (Hu et al., 2022) is proposed to model the weight changes for each layer during fine-tuning. It effectively decreases the number of trainable parameters, thereby lowering the memory burden for training and storage. This approach is currently the mainstream because it avoids adding overhead during inference while demonstrating strong performance (Wang et al., 2023).

Many works have been proposed to enhance the performance of LoRA. AdaLoRA (Zhang et al., 2023a) dynamically prunes insignificant weights during fine-tuning through singular value decomposition (SVD), enabling allocating more rank to important areas under a fixed parameter budget. DoRA (Liu et al., 2024) improves optimization performance by decomposing weight updates into their direction and magnitude components. LoRA+ (Hayou et al., 2024) proposes to use different learning rates for the two matrices in LoRA to improve convergence. PiSSA (Meng et al., 2024) proposes to use the SVD decomposition of the original matrix $W$ to initialize the LoRA matrices, which provides a better initialization for LoRA parameters. LoRA-GA (Wang et al., 2024) proposes to align the gradient of LoRA to that of full fine-tuning at initialization. LoRA-Pro (Wang & Liang, 2025) further proposes to align each gradient step to the full fine-tuning. Li et al. (2024a) develop a resource-efficient SAM variant, called Balancedness-Aware Regularization (BAR), tailored for scale-invariant problems, such as LoRA optimization. In this paper, we improve LoRA by optimizing the sharpness of the loss landscape in the full parameter space, and our approach is orthogonal to previous works.

## 3. Method

In this section, we first briefly review Low-Rank Adaptation (LoRA). Then, we introduce our LoRA optimization objective that considers the landscape flatness of the full parameter space. Finally, we describe our random perturbation generation strategy for effectively improving generalization.

### 3.1. LoRA: Low-Rank Adaptation

Based on the finding that DNNs' optimization happens in a subspace with a much smaller dimension than the number of parameters (Li et al., 2018a; 2022a), LoRA utilizes low-rank matrices to model the change for each layer's weights $W \in \mathbb{R}^{m \times n}$ during fine-tuning as $\Delta W = BA$, where

$B \in \mathbb{R}^{m \times r}$ and $A \in \mathbb{R}^{r \times n}$ with rank $r \ll \min\{m, n\}$ for parameter efficiency. We omit the scaling factor $s = \alpha/r$ here for simplicity, as it can be merged into $A$ and $B$. For the original output $h = Wx$, the modified forward pass is

$$h = Wx + \Delta Wx = (W + BA)x. \quad (1)$$

At initialization, matrix $A$ is commonly drawn from Kaiming distribution (He et al., 2015) and matrix $B$ is set to zeros. During the training, only the low-rank matrices $A$ and $B$ are optimized with the pre-trained weight $W$ being frozen. During the inference, the low-rank matrices $\Delta W$ are merged to the pre-trained weight $W$, and there are no additional computation or memory costs.

### 3.2. LoRA with a Flat Landscape

Despite recent efforts to improve LoRA performance, most studies focus solely on finding solutions performing well in the LoRA optimization space, specifically the rank-$r$ matrix space $\mathcal{M}_r = \{\Delta W \in \mathbb{R}^{m \times n} \mid \text{rank}(\Delta W) \leq r\}$ (focusing on a single LoRA module). Following the well-established sharpness-aware minimization (SAM) objective (Foret et al., 2020), a natural approach is to apply SAM to LoRA parameters (LoRA-SAM) (Li et al., 2024a) with:

$$\min_{A,B} \max_{\|(\varepsilon_A, \varepsilon_B)\| \leq \rho} L\left(W + (B + \varepsilon_B)(A + \varepsilon_A)\right), \quad (2)$$

where $L(\cdot)$ denotes the loss objective for a specific task, $\varepsilon_B \in \mathbb{R}^{m \times r}, \varepsilon_A \in \mathbb{R}^{r \times n}$ are the adversarial weight perturbations over low-rank matrices, $\|(\varepsilon_A, \varepsilon_B)\|$ denotes the total norm of weight perturbations (typically using the $\ell_2$-norm), and $\rho$ is the neighborhood radius.

However, focusing solely on the properties of the optimization space defined by LoRA parameters may have limitations. During inference, the low-rank adaptation $\Delta W$ is merged into the pre-trained weights $W$. A solution that performs well within the LoRA space may be situated in a sharp region of the full parameter space, as illustrated in Figure 1, which could potentially harm overall generalization. To be more clear, employing first-order Taylor expansion for approximation to solve the inner maximum problem in Eqn. (2) (Foret et al., 2020), the equivalent weight perturbation applied to $W$ by Eqn. (2) is

$$\begin{aligned} \varepsilon_W &= B\varepsilon_A + \varepsilon_B A + \varepsilon_B \varepsilon_A \\ &= c\left[BB^\top(\nabla_W L) + (\nabla_W L)A^\top A\right] \\ &\quad + c^2(\nabla_W L)A^\top B^\top(\nabla_W L), \end{aligned} \quad (3)$$

where $\nabla_W L$ is the gradient w.r.t. full parameter weights $W$ and $c = \rho/\sqrt{\|B^\top(\nabla_W L)\|_F^2 + \|(\nabla_W L)A^\top\|_F^2}$ is a scaling factor, with $\|\cdot\|_F$ denoting the Frobenius norm.

Notably, when $B$ is initialized as zero as defaulted in Hu et al. (2022), $B$ will remain small during the training (Hao

et al., 2024) and Eqn. (3) roughly becomes:

$$\varepsilon_W \approx c\left(\nabla_W L\right)A^\top A. \qquad (4)$$

We also empirically validate this in Appendix B. Eqn. (4) indicates that LoRA-SAM only optimizes sharpness within the column space spanned by $A$, which constitutes a small subspace of the full parameter space. As demonstrated in Table 6, applying SAM's sharpness optimization exclusively to LoRA parameters compromises generalization improvements compared to applying it to the full parameter space.

Therefore, it is crucial to consider the loss landscape in the full parameter space and identify a low-rank adaptation that positions the merged weights in a flat region. To achieve this goal, we propose the following flat loss objective:

$$\min_{A,B} \max_{\|\varepsilon_W\|_F \leq \rho} L(W + BA + \varepsilon_W), \qquad (5)$$

where $\varepsilon_W \in \mathbb{R}^{m \times n}$ is the adversarial perturbation over the full parameters. However, directly applying SAM to optimize the sharpness of the full weight space has several disadvantages: 1) it doubles the training cost, which is less desirable for large models, and 2) it requires storing an additional copy of weights for restoring perturbation, which contradicts the principle of parameter-efficient fine-tuning.

To achieve a flatter loss landscape while maintaining time and memory efficiency, we propose relaxing the maximization problem in Eq. (5) to an expectation, resulting in the following Bayesian expected loss objective:

$$\min_{A,B} \quad \mathbb{E}_{(\varepsilon_W)_{i,j} \sim \mathcal{N}(0,\sigma^2)} \quad L(W + BA + \varepsilon_W), \quad (6)$$

where $\sigma^2$ denotes the noise variance, which will be further discussed in Section 3.3. This expected loss smooths the loss function in the full parameter space, as shown in the following lemma, promoting convergence to flatter minima.

**Lemma 3.1** (Bisla et al.). *Assume the loss function $L(W)$ is $\alpha$-Lipschitz continuous and $\beta$-smooth w.r.t. $W$ under $\ell_2$-norm. The smoothed function $\mathbb{E}_{(\varepsilon_W)_{i,j} \sim \mathcal{N}(0,\sigma^2)} L(W + \varepsilon_W)$ is $\min\left\{\frac{\alpha}{\sigma}, \beta\right\}$-smooth w.r.t. $W$.*

To optimize Eqn. (6), we sample a noise matrix $\varepsilon_W$ for each optimization step and compute the perturbed gradient to optimize the low-rank matrices $A$ and $B$. Note that the noise perturbation, generated based on merged model weights, eliminates the need for additional gradient steps required by SAM. In practice, we recommend gradually increasing the perturbation strength to progressively recover flatter minima for better performance.

### 3.3. Effective Random Perturbation Generation

In this section, we introduce our approach for generating random weight perturbations aimed at optimizing sharpness

and improving generalization. Let $W' = W + BA$ denote the merged weight matrix $W' \in \mathbb{R}^{m \times n}$ for a linear layer with input dimension $n$ and output dimension $m$. Our design considers the following two key aspects:

- Filter structure: We aim to generate the weight perturbation by filter (Bisla et al., 2022). There are $m$ filters $W' = (W'_{1,:}, W'_{2,:}, \cdots, W'_{m,:})$ that process the input $x \in \mathbb{R}^n$. Elements within a filter of a larger norm should receive a larger strength of perturbation.

- Input dimension: To ensure that the variance introduced during the forward pass by random weight perturbation is independent of the input dimension, we scale the variance of noise added to each element by a factor of $1/n$, where $n$ is the input dimension.

Our random weight generation scheme is formulated as:

$$(\varepsilon_W)_{i,j} \sim \mathcal{N}\left(0, \frac{\sigma^2}{n}\|W'_{i,:}\|_2^2\right), \qquad (7)$$

where $\sigma$ is a hyper-parameter that controls the perturbation strength. Figure 2 illustrates the comparison between LoRA and Flat-LoRA.

We then analyze the effects of introducing random weight perturbation on the activation. Given an input $x \in \mathbb{R}^n$, and under the hypothesis that $x$ is a random vector where each element has the same variance $\text{var}[x_i]$ and expectation $\mathbb{E}[x_i]$, we have:

$$\text{var}[W'_{j,:}x] = \|W'_{j,:}\|_2^2 \cdot \text{var}[x_i]. \qquad (8)$$

After injecting random weight perturbation $\varepsilon$, we have:

$$\begin{aligned}
&\text{var}\left[(W' + \varepsilon_W)_{j,:}\, x\right] \\
&= \|W'_{j,:}\|_2^2 \cdot \text{var}[x_i] + \sum_{i=1}^{n} \text{var}\left[\varepsilon_{W_{j,i}} x_i\right] \\
&= \|W'_{j,:}\|_2^2 \cdot \text{var}[x_i] + n \cdot \frac{\sigma^2}{n}\|W'_{j,:}\|_2^2 \cdot \left(\text{var}[x_i] + \mathbb{E}^2[x_i]\right) \\
&= (1 + \sigma^2)\|W'_{j,:}\|_2^2 \cdot \text{var}[x_i] + \sigma^2\|W'_{j,:}\|_2^2 \cdot \mathbb{E}^2[x_i]. \quad (9)
\end{aligned}$$

The injection of random weight perturbations $\varepsilon_W$ increases the forward activation variance by a factor of $1 + \sigma^2$, along with a bias term determined by $\mathbb{E}[x_i]$. This amplified variance facilitates escape from sharp local minima. By incorporating a scaling factor $1/n$ in the noise generation process, the variance increase becomes independent of input dimension $n$, as formalized in the following:

**Proposition 3.2.** *For input $x \in \mathbb{R}^n$ with identical variance and mean across elements, injecting random weight perturbations according to Eqn. (7) increases the output variance independently of the input dimension $n$.*

Additionally, we note that this variance would not increase exponentially during the forward propagation of the network due to the existence of layer normalization.

**Storing random seed for memory efficiency.** Memory cost is crucial for parameter-efficient fine-tuning. Optimizing Eqn. (6) requires generating random perturbation $\varepsilon_W$ and computing gradient $\nabla_W L(W + BA + \varepsilon_W)$. While storing the full weight perturbation for large models would be prohibitive, it is sufficient to store only the seed for the random generator and filter norms $\left\{\|W'_{1,:}\|_2^2, \|W'_{2,:}\|_2^2, \cdots, \|W'_{m,:}\|_2^2\right\}$. This allows for the reconstruction of $\varepsilon_W$ when needed. This approach requires minimal memory overhead (i.e., $\mathcal{O}(m)$), in contrast to SAM, which requires storing a full perturbation copy ($\mathcal{O}(m \times n)$) when optimizing sharpness in the full parameter space.

**Simple approach for mixed precision training.** Mixed-precision training, common in large-scale applications, enables memory-efficient integration of perturbation injection during precision casting. Since this training mode maintains both FP32 and FP/BF16 weight copies, we can inject perturbations during the half-precision auto-cast step before forward propagation, eliminating the need to store perturbations or filter norms. However, our primary approach—storing perturbations via filter norms and random seeds—remains more versatile as it functions independently of mixed-precision training.

# 4. Experiments

In this section, we evaluate the performance of Flat-LoRA on diverse tasks: natural language understanding, image classification, dialogue generation, mathematical reasoning, coding abilities, and text-to-image generation. We then demonstrate its enhanced out-of-domain generalization ability, followed by ablation studies and discussions. The code is provided in supplementary materials.

## 4.1. Natural Language Understanding

**Setting.** We fine-tune the T5-Base model on several datasets from the GLUE benchmark, including MNLI, SST, CoLA, QNLI, and MRPC, following Wang et al. (2024). Performance is evaluated on the development set using accuracy as the primary metric. We use LoRA with rank 8 and LoRA alpha 16. We fine-tune the models with 10 epochs using a cosine learning rate schedule; except for MNLI and QNLI, we use 1 epoch. We use a learning rate of 0.0005 for LoRA fine-tuning and 0.0001 for full fine-tuning. The random perturbation strength $\sigma$ is set to 0.05 with a cosine-increasing strategy. Mean and standard deviation are calculated over 3 independent trials.

**Results.** As shown in Table 1, Flat-LoRA consistently outperforms LoRA at ranks 8 and 16, achieving average performance gains of 0.48% and 0.57%, respectively. The improvements are particularly notable on smaller datasets, such as CoLA and MRPC, with gains of 1.19% and 0.94%, respectively, at rank 16. This is because smaller datasets are more prone to overfitting, and Flat-LoRA effectively mitigates this issue, leading to greater performance improvements compared to LoRA.

## 4.2. Image Classification

**Setting.** We fine-tune the CLIP ViT-B/32 model on five image classification tasks, including CIFAR-10/100 (Krizhevsky & Hinton, 2009), Cars (Krause et al., 2013), SVHN (Netzer et al., 2011), and DTD (Cimpoi et al., 2014). We resize all input images to a size of $224 \times 224$ and freeze the classification head. We experiment with LoRA using ranks of 8 and 16 and fine-tune the models for 10 epochs under a cosine annealing schedule. The learning rate is set to 0.0005 for LoRA and Flat-LoRA and 0.0001 for full fine-tuning, with a weight decay of 0.1. The perturbation strength $\sigma$ is set to 0.15 for Flat-LoRA with a cosine-increasing strategy. The mean and standard deviation are calculated over 3 independent trials.

**Results.** We measure the performance with classification accuracy and report the results in Table 2. Again, we observe that Flat-LoRA significantly outperforms LoRA at both ranks 8 and 16, achieving averaged improvements of 0.56% and 0.74%, respectively. Notably, Flat-LoRA with rank 8 surpasses both LoRA with rank 16 and full fine-tuning by 0.28%. These results confirm the effectiveness of optimizing the loss landscape's sharpness in the full parameter space.

## 4.3. Large Language Model

**Setting.** To evaluate the scalability of Flat-LoRA, we further conduct experiments on large language models. Specifically, we fine-tune Llama 2-7B (Touvron et al., 2023) on three tasks: *chat*, *math*, and *code*, following Wang et al. (2024). We use a learning rate of 5e-4 and employ a cosine learning rate scheduler with a warmup ratio of 0.03. The LoRA rank is set to 8 with LoRA alpha 16, and the training epoch is set to 2. The backbone uses BF16 precision, with the parameters of LoRA modules set to FP32 precision. For *chat* task, we fine-tune the model on WizardLM (Xu et al., 2023) and test on the MT-Bench dataset (Zheng et al., 2023). For *math* task, we fine-tune the model on MetaMathQA (Yu et al., 2024) and evaluate it on GSM8K evaluation set (Cobbe et al., 2021). For *code* task, we fine-tune the model on Code-Feedback (Zheng et al., 2024) and evaluate it on HumanEval (Chen et al., 2021). Training uses 52K for chat and 100K samples for math and code tasks. The random perturbation strength $\sigma$ is set to 0.05 with a cosine-increasing strategy.

**Results.** We measure the performance of the *chat* task by the first-turn score with GPT-4, the *math* task by accuracy,

Table 1: Results (%) on fine-tuning T5-base with a subset of GLUE datasets.

| Method | MNLI | SST2 | CoLA | QNLI | MRPC | Avg. |
|---|---|---|---|---|---|---|
| Full FT | $86.19_{\pm0.04}$ | $94.15_{\pm0.09}$ | $82.84_{\pm0.12}$ | $93.10_{\pm0.04}$ | $89.22_{\pm0.23}$ | 89.10 |
| LoRA ($r=8$) | $\mathbf{86.24}_{\pm0.02}$ | $94.25_{\pm0.07}$ | $82.87_{\pm0.22}$ | $93.06_{\pm0.03}$ | $88.56_{\pm0.37}$ | 88.99 |
| Flat-LoRA ($r=8$) | $86.20_{\pm0.04}$ | $\mathbf{94.75}_{\pm0.20}$ | $\mathbf{83.61}_{\pm0.38}$ | $\mathbf{93.16}_{\pm0.09}$ | $\mathbf{89.59}_{\pm0.37}$ | $\mathbf{89.47}$ |
| LoRA ($r=16$) | $86.49_{\pm0.06}$ | $94.52_{\pm0.21}$ | $82.89_{\pm0.44}$ | $92.97_{\pm0.05}$ | $88.89_{\pm0.44}$ | 89.15 |
| Flat-LoRA ($r=16$) | $\mathbf{86.51}_{\pm0.01}$ | $\mathbf{94.84}_{\pm0.02}$ | $\mathbf{84.08}_{\pm0.31}$ | $\mathbf{93.28}_{\pm0.03}$ | $\mathbf{89.83}_{\pm0.34}$ | $\mathbf{89.72}$ |

Table 2: Results (%) on fine-tuning CLIP ViT-B/32 with image classification datasets.

| Method | CIFAR-10 | CIFAR-100 | Cars | SVHN | DTD | Avg. |
|---|---|---|---|---|---|---|
| Full FT | $97.99_{\pm0.01}$ | $89.06_{\pm0.11}$ | $73.30_{\pm0.43}$ | $97.44_{\pm0.03}$ | $76.80_{\pm0.25}$ | 86.92 |
| LoRA ($r=8$) | $97.90_{\pm0.02}$ | $87.74_{\pm0.13}$ | $73.22_{\pm0.53}$ | $97.49_{\pm0.08}$ | $76.86_{\pm0.34}$ | 86.64 |
| Flat-LoRA ($r=8$) | $\mathbf{98.09}_{\pm0.04}$ | $\mathbf{88.64}_{\pm0.23}$ | $\mathbf{74.17}_{\pm0.71}$ | $\mathbf{97.59}_{\pm0.04}$ | $\mathbf{77.51}_{\pm0.28}$ | $\mathbf{87.20}$ |
| LoRA ($r=16$) | $97.99_{\pm0.03}$ | $88.12_{\pm0.23}$ | $73.80_{\pm0.42}$ | $97.56_{\pm0.08}$ | $77.34_{\pm0.32}$ | 86.92 |
| Flat-LoRA ($r=16$) | $\mathbf{98.21}_{\pm0.04}$ | $\mathbf{89.27}_{\pm0.07}$ | $\mathbf{74.89}_{\pm0.52}$ | $\mathbf{97.71}_{\pm0.10}$ | $\mathbf{78.24}_{\pm0.44}$ | $\mathbf{87.66}$ |

and the *code* task by PASS@1 metric. From the results in Table 3, we observe that Flat-LoRA significantly enhances LoRA's performance, achieving an improvement of +0.20 on the MT-Bench dataset, +3.18% on the GSM8K dataset, and +3.08% on the Human-Eval dataset. Notably, these gains are substantially larger than those observed on smaller models, such as T5-base and CLIP ViT-B/32, highlighting the significance of pursuing flat minima for large-scale models. Moreover, the baselines we adopted are considerably stronger than those reported in previous works; for instance, we achieve 57.47% (ours) versus 42.08% (Wang et al. (2024)) for LoRA on the GSM8K dataset. Despite these stronger baselines, Flat-LoRA continues to deliver significant accuracy improvements over the standard LoRA, demonstrating its effectiveness in enhancing generalization.

Table 3: Results on fine-tuning Llama 2-7B.

| Method | MT-Bench | GSM8K | Human-Eval |
|---|---|---|---|
| Full FT | $5.30_{\pm0.11}$ | $59.36_{\pm0.85}$ | $35.31_{\pm2.13}$ |
| LoRA ($r=8$) | $5.96_{\pm0.03}$ | $57.47_{\pm0.45}$ | $24.85_{\pm0.52}$ |
| Flat-LoRA ($r=8$) | $\mathbf{6.16}_{\pm0.05}$ | $\mathbf{60.65}_{\pm0.63}$ | $\mathbf{27.93}_{\pm0.79}$ |

## 4.4. Text-to-Image Generation

**Setting.** We fine-tune the SDXL model (Podell et al., 2023) with the pipeline of Dreambooth (Ruiz et al., 2023) and the scripts implemented by HuggingFace. The finetuning dataset, 3D Icons[1], contains 23 training images, all of which have a square. We fine-tune the model for 500 steps with

a constant learning rate of 0.0001. The batch size is set to 1. The LoRA rank and alpha are set to 4. The random perturbation strength $\sigma$ is set to 0.1 for Flat-LoRA. Other hyperparameters are set to default values.

**Results.** As shown in Figure 3, Flat-LoRA exhibits better personalization than LoRA while maintaining better generation ability. For instance, in the second column, the image generated by Flat-LoRA includes a distinctive square behind the bird, aligning more closely with the "icon" feature present in the training images (top row). Furthermore, Flat-LoRA more effectively preserves the concept of eyes, whereas, in columns 1, 3, and 5, the birds generated by LoRA are missing eyes.

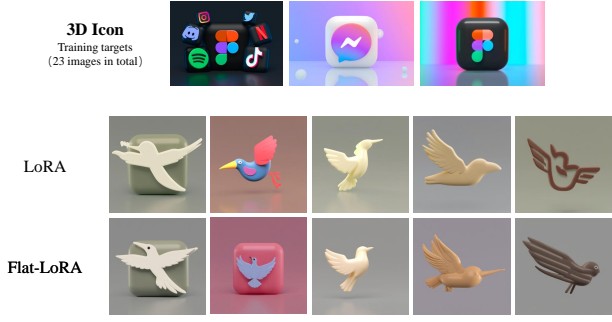

Prompt: a TOK icon of a flying bird, in the style of TOK

Figure 3: Images generated by SDXL fine-tuned with LoRA and Flat-LoRA on 3D icon datasets. Each column uses the *same* seeds for fair comparison.

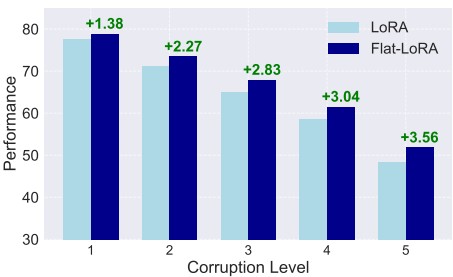

Figure 4: Performance comparison of LoRA and Flat-LoRA across different corruption levels of CIFAR-100-C. The model is fine-tuned on CIFAR-100 with CLIP ViT-B/32.

## 4.5. Out-of-Domain Generalization

Flat minima have been shown to better accommodate distributional shifts between training and test data, thereby improving out-of-domain generalization. This property is particularly critical for pretrained vision and language models, which are designed for a wide range of applications. Below, we explore this property of Flat-LoRA in detail.

**Corruption datasets.** We focus on image classification tasks to evaluate the robustness of Flat-LoRA under data distribution shifts. Specifically, we fine-tune CLIP ViT-B/32 on CIFAR-100 and evaluate the model on corrupted CIFAR-100-C (Hendrycks & Dietterich, 2019). The results across varying levels of corruption severity are presented in Figure 4. Flat-LoRA consistently outperforms LoRA, with performance gains increasing as corruption severity rises, from +1.38% at level 1 to +3.56% at level 5. These results demonstrate that the flatter minima identified by Flat-LoRA enhance out-of-domain generalization compared to LoRA.

**Instruction following.** We fine-tune the Llama 2-13B model on the Alpaca dataset (Taori et al., 2023), which simulates real-world variability and prepares the model to handle unseen or shifted distributions at test time. The model is evaluated on InstructEval (Chia et al., 2023), an instruction-following benchmark, using the official code provided by Chia et al. (2023). The experimental setup follows Ren et al. (2024). From the results in Table 4, we observe that Flat-LoRA consistently outperforms LoRA. Notably, improvements on DROP and Human-Eval are more pronounced (+0.71% and +1.83%, respectively).

Table 4: Results on instruct-following tasks. We fine-tune the Llama 2-13B model on the Alpaca datasets and evaluate the performance using the InstructEval metrics.

| Method | MMLU | DROP | BBH | Human-Eval |
|---|---|---|---|---|
| Full FT | $52.36_{\pm 0.45}$ | $38.23_{\pm 0.47}$ | $35.38_{\pm 0.35}$ | $15.44_{\pm 0.35}$ |
| LoRA ($r = 8$) | $51.22_{\pm 0.38}$ | $37.26_{\pm 0.63}$ | $34.77_{\pm 0.22}$ | $13.01_{\pm 0.93}$ |
| Flat-LoRA ($r = 8$) | $\mathbf{51.88}_{\pm 0.55}$ | $\mathbf{38.18}_{\pm 0.71}$ | $\mathbf{35.22}_{\pm 0.26}$ | $\mathbf{15.24}_{\pm 0.61}$ |

Table 5: Comparison with other LoRA variants. The experiments are conducted on the GLUE subsets using the T5-Base model.

| Method | Dataset | |
|---|---|---|
| | CoLA | MRPC |
| *Baseline Methods* | | |
| LoRA (Hu et al., 2022) | $82.87_{\pm 0.22}$ | $88.56_{\pm 0.37}$ |
| PiSSA (Meng et al., 2024) | $83.18_{\pm 0.24}$ | $88.96_{\pm 0.44}$ |
| LoRA-GA (Wang et al., 2024) | $83.13_{\pm 0.45}$ | $88.73_{\pm 0.48}$ |
| DoRA (Liu et al., 2024) | $83.16_{\pm 0.15}$ | $89.46_{\pm 0.37}$ |
| AdaLoRA (Zhang et al., 2023b) | $82.58_{\pm 0.56}$ | $88.79_{\pm 0.33}$ |
| LoRA+ (Hayou et al., 2024) | $82.65_{\pm 0.23}$ | $89.30_{\pm 0.47}$ |
| *Our Methods* | | |
| Flat-LoRA | $\mathbf{83.61}_{\pm 0.38}$ | $89.59_{\pm 0.37}$ |
| Flat-PiSSA | $83.51_{\pm 0.48}$ | $89.89_{\pm 0.71}$ |
| Flat-LoRA-GA | $83.41_{\pm 0.45}$ | $89.20_{\pm 0.49}$ |
| Flat-DoRA | $83.56_{\pm 0.27}$ | $\mathbf{89.99}_{\pm 0.47}$ |
| Flat-AdaLoRA | $83.13_{\pm 0.28}$ | $89.23_{\pm 0.34}$ |
| Flat-LoRA+ | $83.56_{\pm 0.46}$ | $89.61_{\pm 0.44}$ |

## 4.6. Integration with Other Methods

In this section, we compare Flat-LoRA with recently proposed LoRA variants, including PiSSA, LoRA-GA, DoRA, AdaLoRA, and LoRA+. Experiments are conducted on the CoLA and MRPC datasets using the T5-base model with LoRA rank 8. The results are presented in Table 5. We observe that Flat-LoRA consistently outperforms previous methods on both datasets by 0.53% and 0.13%, respectively. Furthermore, the flat loss objective can be seamlessly integrated with the previous approaches to yield consistent improvements on both datasets by 0.91% and 0.93%, respectively. Note that these improvements are achieved at minimal additional cost, as shown in Table 7. This highlights the scalability of our approach and the effectiveness of considering the sharpness of the full parameter space.

## 4.7. Ablation Studies and Discussion

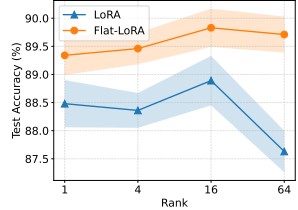 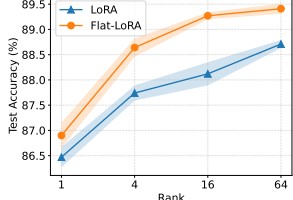

(a) MRPC with T5-Base     (b) CIFAR-100 with ViT-B/32

Figure 5: Performance comparison across different LoRA ranks. Keeping the LoRA alpha fixed at 16, we vary the LoRA ranks among $\{1, 4, 16, 64\}$. The results are averaged over three independent trials.

**Results under different LoRA ranks.** Following the settings in Section 4.1 and 4.2, we evaluate the performance

Table 6: Comparison with SAM on the GLUE subsets using the T5-Base model.

| Method | Flat Space | CoLA | MRPC | Additional Memory | Training time |
|--------|-----------|------|------|-------------------|---------------|
| LoRA | - | $82.87_{\pm0.59}$ | $88.56_{\pm0.37}$ | - | $1\times$ |
| LoRA+SAM | $A, B$ | $83.31_{\pm0.48}$ | $88.98_{\pm0.22}$ | $\mathcal{O}((m+n)\times r)$ | $2\times$ |
| LoRA+SAM | $W$ | $\mathbf{83.67}_{\pm0.39}$ | $89.26_{\pm0.32}$ | $\mathcal{O}(m\times n)$ | $2\times$ |
| Flat-LoRA | $A, B$ | $83.19_{\pm0.70}$ | $88.81_{\pm0.51}$ | $\mathcal{O}(m+r)$ | $1\times$ |
| Flat-LoRA | $W$ | $83.61_{\pm0.38}$ | $\mathbf{89.59}_{\pm0.37}$ | $\mathcal{O}(m)$ | $1\times$ |

of Flat-LoRA under different LoRA ranks. The results are shown in Figure 5. We observe that Flat-LoRA consistently outperforms LoRA across different LoRA ranks by +1.10% on MRPC and +1.15% on CIFAR-100. Even at LoRA rank 1, which is typically underfitting, Flat-LoRA still delivers a significant performance boost over LoRA. This highlights the importance of considering the sharpness of the full parameter space. Additionally, as the LoRA rank increases, we observe that LoRA's performance can degrade due to overfitting, particularly on MRPC, which is a small dataset with 3.7k data points. Flat-LoRA effectively mitigates this overfitting issue by identifying flatter minima that generalize better. Thus, we conclude that Flat-LoRA enhances LoRA fine-tuning performance not only in underfitting scenarios, where the rank is low and limited information from the full parameter space is explored, but also in high LoRA rank situations, where the risk of overfitting is more pronounced.

**Comparison with SAM.** We compare Flat-LoRA to SAM integrated with LoRA across different flat spaces: applying SAM's sharpness optimization to the full parameter space ($W$) and to the LoRA parameters ($A, B$). Following the setup described in Section 4.1, we evaluate perturbation radii $\rho$ over $\{0.001, 0.003, 0.005, 0.01, 0.05, 0.1, 0.2, 0.5\}$, finding that $\rho = 0.05$ yields optimal performance when applied to the full parameter space ($W$), while $\rho = 0.003$ is optimal for the LoRA parameters ($A, B$). From the results in Table 6, we observe that applying SAM to the full parameter space ($W$) consistently outperforms its application to the LoRA parameters ($A, B$), achieving improvements of +0.36% on CoLA and +0.28% on MRPC. However, SAM over $W$ incurs an additional memory overhead of $\mathcal{O}(m\times n)$ to store adversarial weight perturbations, rendering it impractical for parameter-efficient training. By contrast, Flat-LoRA achieves performance comparable to, or better than, SAM applied to $W$, while requiring only $\mathcal{O}(m)$ additional memory. Furthermore, Flat-LoRA preserves the training efficiency of vanilla LoRA ($1\times$), whereas SAM-based approaches double the training time ($2\times$) due to the need for additional gradient computations.

**Memory and time costs.** In Table 7, we report the memory and time usage for fine-tuning MetaMathQA datasets using the Llama 2-7B model. The training settings are the same

with Section 4.3, and we use a micro-batch size of 2, running on an NVIDIA GeForce RTX 4090 GPU. Flat-LoRA is implemented based on our default random seed approach. We observe that Flat-LoRA adds minimal overhead compared to LoRA - only 0.12GB of extra memory and 11 minutes of training time. These results highlight that Flat-LoRA can be conveniently integrated into LoRA training with little additional overhead.

Table 7: Comparison of memory and time usage

| Method | Memory | Training Time | GSM8K (%) |
|--------|--------|---------------|-----------|
| LoRA | 23.49GB | 7h 22min | $57.47_{\pm0.45}$ |
| Flat-LoRA | 23.61GB | 7h 33min | $60.65_{\pm0.63}$ |

**Landscape visualization.** In Figure 6, we visualize the surfaces of the loss landscape for LoRA and Flat-LoRA at ranks 1 and 16. Following the technique proposed by Li et al. (2018b), we plot the loss surface along random "filter-normalized" directions in the full parameter space ($W$). For both LoRA and Flat-LoRA, the merged weights are used for visualization. The results demonstrate that Flat-LoRA consistently achieves a significantly flatter loss landscape compared to LoRA at both ranks. Notably, when the LoRA rank is lower, the corresponding loss landscape tends to be sharper, highlighting the importance of optimizing the sharpness in the full parameter space.

## 5. Conclusion

We present Flat-LoRA, an efficient low-rank adaptation method that optimizes the sharpness of the loss landscape in the full parameter space. Unlike conventional sharpness-aware minimization approaches that impose heavy computation and memory overhead, we employ the Bayesian expectation loss objective to pursue flat minima and design refined generation schemes for random weight perturbations while maintaining efficiency. Extensive experiments across natural language processing and computer vision demonstrate Flat-LoRA's effectiveness in improving both in-domain and out-of-domain generalization.

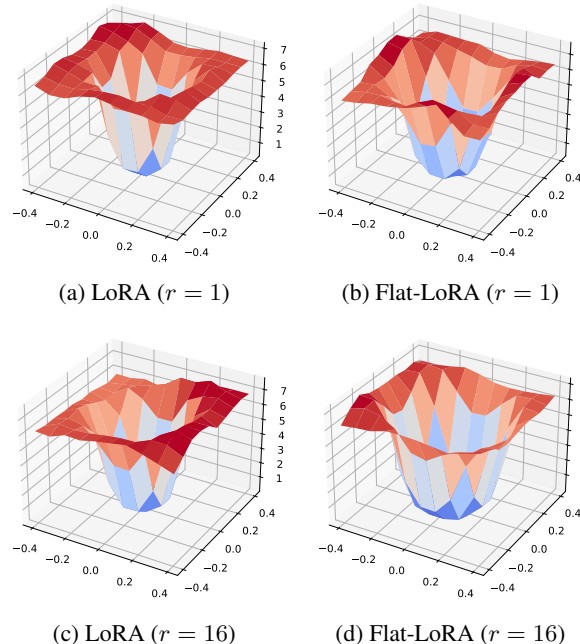

(a) LoRA ($r = 1$)  (b) Flat-LoRA ($r = 1$)

(c) LoRA ($r = 16$)  (d) Flat-LoRA ($r = 16$)

Figure 6: Loss landscape visualization in the full parameter space. The experiments are conducted on CIFAR-100 with CLIP ViT-B/32.

## Impact Statement

This paper presents work whose goal is to advance the field of Machine Learning. There are many potential societal consequences of our work, none which we feel must be specifically highlighted here.

## Acknowledgment

This work was supported by National Key Research Development Project (2023YFF1104202) and National Natural Science Foundation of China (62376155).

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

## A. Training-vs-Test Loss and Generalization Gap Curves

We plot the training-vs-test loss curves and generalization gap on CIFAR-100 and MRPC datasets in Figure A1. The results show Flat-LoRA exhibits slightly higher training loss than LoRA, with a smaller generalization gap between training and test accuracies. Thus, we can conclude that the gains of Flat-LoRA are not due to lower training loss but due to better optimization that confers better generalization.

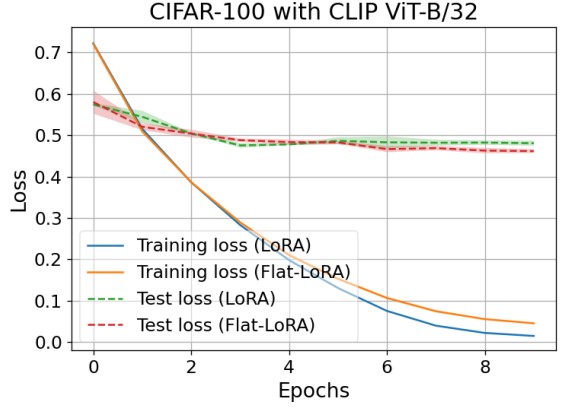

(a) Training/test loss curves on CIFAR-100.

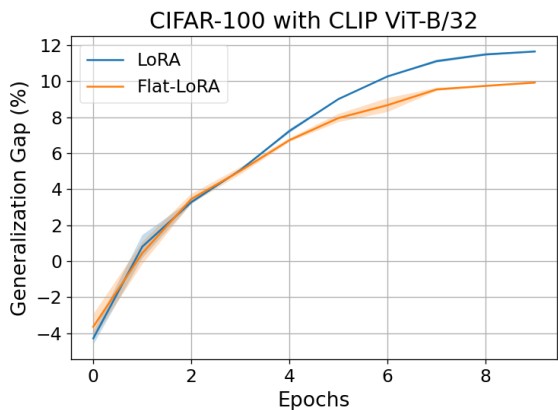

(b) Generalization gap curves on CIFAR-100.

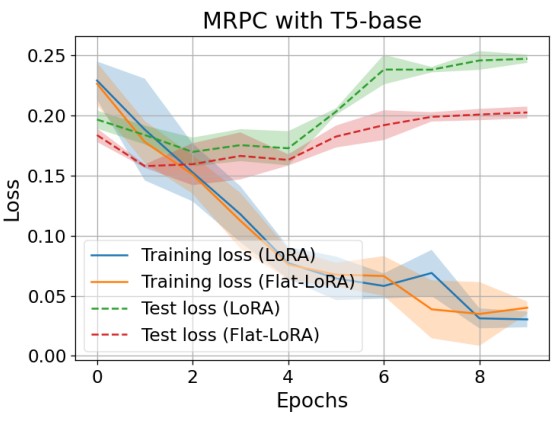

(c) Training/test loss curves on MRPC.

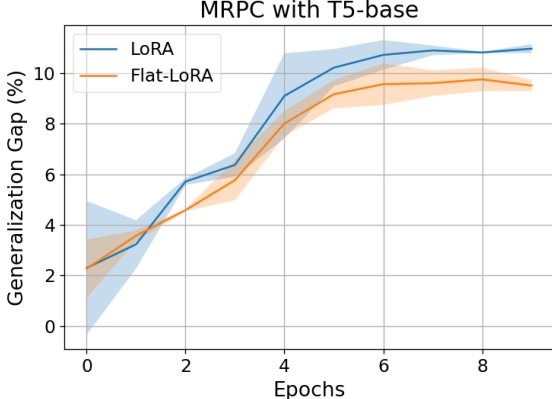

(d) Generalization gap curves on MRPC.

Figure A1: Training-vs-test loss and generalization gap curves comparison. Flat-LoRA exhibits slightly higher training loss than LoRA, with a smaller generalization gap between training and test accuracies.

## B. Validation on the Components of $\varepsilon_W$

In this section, we validate the approximation of Eqn. (4), i.e., $\varepsilon_W \approx \varepsilon_B A = c(\nabla_W L)A^\top A$. We conduct an experiment on the MRPC dataset with T5-base model and record the statistics of $\frac{\|\varepsilon_B A\|}{\|\varepsilon_W\|}$ during the training. The results are shown in Figure A2. We observe that $\frac{\|\varepsilon_B A\|}{\|\varepsilon_W\|} > 0.95$ throughout the training. This validates the approximation of Eqn. (4).

## C. Extending Perturbation to All Layers

We extend the injection of random weight perturbation to all layers, referred to as "Flat-LoRA (all)". Specifically, we additionally add perturbations to layernorm layers, biases, class embeddings, etc. We generate noise based on the absolute weight $|W|$. From the results in Table A1, we observe that Flat-LoRA (all) indeed improves performance, though the improvement is not as large as Flat-LoRA (Linear) over LoRA.

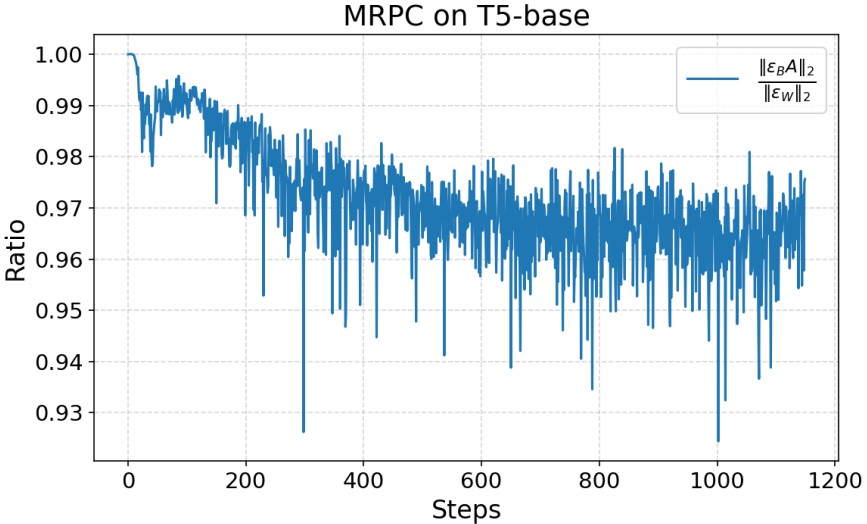

Figure A2: Statistics of $\frac{\|\varepsilon_B A\|_2}{\|\varepsilon_W\|_2}$. We observe that $\frac{\|\varepsilon_B A\|_2}{\|\varepsilon_W\|_2}$ remains almost above 0.95 throughout training, indicating that the actual weight perturbation of LoRA-SAM $\varepsilon_W$ is almost determined by $\varepsilon_B A$. This indicates that LoRA-SAM primarily optimizes the sharpness within the subspace spanned by $A$. The experiment is conducted on the MRPC dataset with the T5-base model.

Table A1: Results on CIFAR-10/100 with CLIP ViT-B/32.

| Method | CIFAR-10 | CIFAR-100 |
|---|---|---|
| LoRA | $97.90_{\pm 0.02}$ | $87.74_{\pm 0.13}$ |
| Flat-LoRA (linear) | $98.09_{\pm 0.04}$ | $88.64_{\pm 0.23}$ |
| Flat-LoRA (all) | $\mathbf{98.13}_{\pm 0.03}$ | $\mathbf{88.76}_{\pm 0.19}$ |

## D. Ablation on the Variance Magnitude

To evaluate the impact of the perturbation variance magnitude $\sigma$ for Flat-LoRA, we vary $\sigma$ among $\{0, 0.01, 0.05, 0.10, 0.15, 0.20\}$ and fine-tune CIFAR-100 on CLIP ViT-B/32 and ViT-L/14 as well as GSM8k on Llama 2-7B and Llama 2-13B. From the results in Table A2 and Table A3, we observe that the optimal results are achieved when $\sigma$ is 0.05 or 0.10 for both datasets and different network sizes. Hence, we suggest $\sigma = 0.05/0.10$ for practice usage.

Table A2: Results on CIFAR-100 with different variance magnitude.

| $\sigma$ | 0 | 0.01 | 0.05 | 0.10 | 0.15 | 0.20 |
|---|---|---|---|---|---|---|
| ViT-B/32 | $87.74_{\pm 0.13}$ | $88.14_{\pm 0.22}$ | $88.37_{\pm 0.41}$ | $\mathbf{88.65}_{\pm 0.35}$ | $88.64_{\pm 0.23}$ | $88.06_{\pm 0.31}$ |
| ViT-L/14 | $92.13_{\pm 0.17}$ | $92.33_{\pm 0.07}$ | $92.63_{\pm 0.11}$ | $\mathbf{93.11}_{\pm 0.13}$ | $92.98_{\pm 0.21}$ | $92.46_{\pm 0.03}$ |

Table A3: Results on GSM8k with different variance magnitude.

| $\sigma$ | 0 | 0.01 | 0.05 | 0.10 | 0.15 | 0.20 |
|---|---|---|---|---|---|---|
| LLama 2-7B | $57.47_{\pm 0.45}$ | $58.35_{\pm 0.42}$ | $\mathbf{60.65}_{\pm 0.63}$ | $60.56_{\pm 0.48}$ | $60.08_{\pm 0.76}$ | $58.50_{\pm 0.85}$ |
| LLama 2-13B | $66.76_{\pm 0.23}$ | $67.02_{\pm 0.67}$ | $67.75_{\pm 0.70}$ | $\mathbf{68.11}_{\pm 0.53}$ | $67.66_{\pm 0.97}$ | $67.34_{\pm 1.17}$ |

## E. More Comparisons to LoRA's Varints

In Table A4, we compare Flat-LoRA with more recently proposed LoRA varints, including oBAR/nBAR (Li et al., 2024a), LoRA-Pro (Wang & Liang, 2025), GaLore (Zhao et al., 2024), and CorDA (Yang et al., 2024). The experiments are conducted on the T5-base model with MRPC and CoLA datasets. We can observe that Flat-LoRA achieves competitive or better performance than those state-of-the-art variants.

Table A4: Performance comparison on MRPC and CoLA.

| Methods | MRPC | CoLA |
|---|---|---|
| oBAR (Li et al., 2024a) | $88.58_{\pm 0.35}$ | $83.07_{\pm 0.87}$ |
| nBAR (Li et al., 2024a) | $88.63_{\pm 0.42}$ | $82.78_{\pm 0.68}$ |
| LoRA-Pro (Wang & Liang, 2025) | $89.23_{\pm 0.33}$ | $83.17_{\pm 0.28}$ |
| GaLore (Zhao et al., 2024) | $88.90_{\pm 1.12}$ | $83.14_{\pm 0.57}$ |
| CorDA (Yang et al., 2024) | $\mathbf{89.76}_{\pm 0.52}$ | $83.38_{\pm 0.47}$ |
| Flat-LoRA (Ours) | $89.59_{\pm 0.37}$ | $\mathbf{83.61}_{\pm 0.38}$ |

