# OpenReview forum: "Flat-LoRA: Low-Rank Adaptation over a Flat Loss Landscape"
_ICML.cc/2025/Conference — ICML 2025 poster_

### Official Review · Reviewer_cWcV · 2025-03-10

**Overall Recommendation:** 3

**Summary:**

The authors present an approach to improving generalization in PEFT. They argue that generalization is strongly correlated with flat minima and that existing empirical approaches are not directly applicable to PEFT. To address this, they propose Flat-LoRA, an extension of SAM that facilitates the discovery of flat minima in the entire parameter space while training within a subspace. By leveraging existing techniques, such as the relaxation of inner optimization problems using random directions and on-the-fly noise generation, the authors demonstrate that Flat-LoRA consistently improves over baseline methods, albeit with a small difference.

**Claims And Evidence:**

The proposed method optimizes using the same framework as SAM and does not provide a mathematical analysis of the algorithm's convergence. However, experimental results demonstrate a clear advantage of the method. Experiments with OOD data show that Flat-LoRA improves generalization, supporting the paper’s claim.

**Essential References Not Discussed:**

Some recent PEFT methods are missing, including GaLore [1], CorDA [2].

[1] GaLore: Memory-Efficient LLM Training by Gradient Low-Rank Projection
[2] CorDA: Context-Oriented Decomposition Adaptation of Large Language Models for Task-Aware Parameter-Efficient Fine-tuning

**Experimental Designs Or Analyses:**

To prove that the proposed method improves generalization, the authors analyze it in multiple experimental setups, following the widely accepted frameworks and using the same metrics and datasets. As the improvement over most of the benchmarks is minute, it can be beneficial to provide statistical analysis to justify the difference, e.g., in Tables 1 and 2. Improvement in T2I task is shown only qualitatively and requires additional analysis using common datasets (i.e., measure CLIP-T/CLIP-I on DreamBench).

**Methods And Evaluation Criteria:**

The paper presents results on common datasets (GLUE, InstructEval, MT-Bench) for NLP tasks as well as small-scale qualitative T2I experiments to provide sufficient evidence for Flat-LoRA’s effectiveness.

**Other Comments Or Suggestions:**

- Lines 169-171 duplicate lines 167-169.

**Other Strengths And Weaknesses:**

Strengths:

- Paper is well-written and easy to read

Weaknesses:

- Table 4 does not provide variance
- Tables 1, 2, and 5 require more analysis to prove that the difference is statistically significant, as most of the improvement of Flat-LoRA is lower than the standard deviation of the metrics.
- Some of the relevant baselines are mentioned but are not used in comparison (Balancedness-Aware Regularization [3], LoRA-Pro [4], GaLore [1], CorDA [2])

[3] Implicit Regularization of Sharpness-Aware Minimization for Scale-Invariant Problems

[4] LoRA-Pro: Are Low-Rank Adapters Properly Optimized?

**Questions For Authors:**

See weaknesses.

**Relation To Broader Scientific Literature:**

The paper references relevant prior work and discusses concurrent approaches. The proposed method complements these existing approaches and can be effectively combined with them.

**Theoretical Claims:**

The mathematical derivations appear correct to the best of my knowledge.

---

> ### Author Rebuttal · Authors · 2025-03-31
>
> Thanks for your detailed and constructive comments. We address your concerns point by point as follows:
>
>  ---
>  **Q1:** Table 4 does not provide variance.
>
>  **A1:** Following your suggestion, we rerun the experiments and report the variance as follows:
>
> | Method  | MMLU  | DROP |  BBH|  Human-Eval|
> | ---     | ---    | --- | --- | --- |
> | Full FT | 52.36±0.45 | 38.23±0.47 | 35.38±0.35 | 15.44±0.35|
> | LoRA    | 51.22±0.38 | 37.26±0.63 | 34.77±0.22 | 13.01±0.93 |
> | Flat-LoRA | 51.88±0.55 | 38.18±0.71 | 35.22±0.26 | 15.24±0.61 |
>
>
>  ---
>  **Q2:** Tables 1, 2, and 5 require more analysis to prove that the difference is statistically significant, as most of the improvement of Flat-LoRA is lower than the standard deviation of the metrics.
>
>  **A2:** Following your suggestion, we conduct statistical analysis using t-test as follows:
>
>  (Table 1) Hypothesis: Flat-LoRA is better than LoRA
>
>  T5-base | MNLI | SST-2 | CoLA | QNLI | MRPC
>  ---|---| --- | --- | --- | ---
>  p (r=8) | 0.186 | 0.038 | 0.043 | 0.186 | 0.027
>  p (r=16) | 0.623 | 0.117 | 0.023 | 0.002 | 0.046
>
>  (Table 2) Hypothesis: Flat-LoRA is better than LoRA
>  ViT-B/32 | CIFAR-10 | CIFAR-100 | Cars | SVHN | DTD
>  ---|---| --- | --- | --- | ---
>  p (r=8) |0.006 | 0.008 | 0.137 | 0.125 | 0.063 |
>  p (r=16) | 0.001 | 0.001 | 0.047 | 0.112 | 0.046 |
>
>  (Table 5) Hypothesis: Flat-X is better than X, where X is base method, can be LoRA, PiSSA, etc.
>  T5-base | LoRA | PiSSA | LoRA-GA | DoRA | AdaLoRA | LoRA+
>  ---|---|---|---|---|---|---
>  CoLA| 0.043|0.36|0.488|0.088|0.202|0.037
>  MRPC| 0.027 | 0.126| 0.301|0.199|0.183|0.313
>
>
> We observe that, in most cases, the improvements are significant (i.e., p < 0.05). However, we acknowledge that in some cases, such as in Table 5, the improvements are relatively modest. This could be attributed to the strong baseline performance of the LoRA variants, which leaves limited room for further improvement. Moreover, we find that on large-scale training, the improvement is more significant as below, showing the scalability of flat-LoRA.
>
> (Table 3) Hypothesis: Flat-LoRA is better than LoRA
>   Llama2-7b | MT-Bench | GSM8k | Human-Eval
>  ---|---|---|---
>  p (r=8) | 0.007 | 0.003 | 0.007
>
>  ---
>  **Q3:** Some of the relevant baselines are mentioned but are not used in comparison (Balancedness-Aware Regularization [3], LoRA-Pro [4], GaLore [1], CorDA [2]).
>
>  **A3:** Thanks for providing the very relevant works. We compare them below and will add the comparision and citations in the revision.
>
> | T5-base   | oBAR [3]       | nBAR [3]       | LoRA-Pro [4] | GaLore [1]     | CorDA [2]      | Flat-LoRA
> |---|---|---|---|---|--- |---
> | MRPC | 88.58±0.35 | 88.63±0.42 | 89.23 $\pm$ 0.33 | 88.90±1.12 | 89.76±0.52 | 89.59±0.37
> | CoLA | 83.07±0.87 | 82.78±0.68 | 83.17 $\pm$ 0.28 | 83.14±0.57 | 83.38±0.47 |83.61±0.38
>
> [1] GaLore: Memory-Efficient LLM Training by Gradient Low-Rank Projection, ICML'24
>
> [2] CorDA: Context-Oriented Decomposition Adaptation of Large Language Models for Task-Aware Parameter-Efficient Fine-tuning, NeurIPS'24
>
> [3] Implicit Regularization of Sharpness-Aware Minimization for Scale-Invariant Problems, NeurIPS'24
>
> [4] LoRA-Pro: Are Low-Rank Adapters Properly Optimized? ICLR'25
>
>  ---
>  **Q4:** Improvement in T2I task is shown only qualitatively and requires additional analysis using common datasets (i.e., measure CLIP-T/CLIP-I on DreamBench).
>
> **A4:** Thanks for your valuable suggestions. Following your suggestion, we finetune an SDXL model under the same setting as Figure 3, using the public dataset provided by DreamBooth, including 30 tasks, where each task is evaluated with 25 prompts and 4 different seeds. The CLIP-I and CLIP-T scores are reported below, which show consistent results with Figure 3. The model with Flat-LoRA exhibits both higher subject fidelity (CLIP-I) and prompt fidelity (CLIP-T). More comparisons of sample images can be found in [Figure 3](https://anonymous.4open.science/r/Flat-LoRA/rebuttal.pdf).
>
> |                     | Real Image | LoRA  | Flat-LoRA |
> | ------------------- | ---------- | ----- | --------- |
> | CLIP-T ($\uparrow$) | -          | 0.299 | 0.311     |
> | CLIP-I ($\uparrow$) | 0.881      | 0.819 | 0.825     |
>
>
> ---
> **Q5:** Lines 169-171 duplicate lines 167-169.
>
> **A5:** Thanks for your careful reading. We will fixed it in the revision.

---

### Official Review · Reviewer_oboy · 2025-03-12

**Overall Recommendation:** 4

**Summary:**

**1. Summary of Contributions**:
The paper introduces **Flat-LoRA**, a novel method for low-rank adaptation (LoRA) that aims to find parameter-efficient fine-tuning solutions residing in a flatter region of the full parameter space. The authors identify that standard LoRA, while efficient, might converge to minima that appear flat within its low-dimensional optimization space but are sharp in the context of the original, full parameter space, potentially hindering generalization. To address this, Flat-LoRA employs a **Bayesian expectation loss objective with carefully designed random weight perturbations** applied to the merged weights (pre-trained weights + LoRA adaptations). This approach smooths the loss landscape in the full parameter space, encouraging convergence to flatter minima without incurring the significant computational and memory overheads associated with sharpness-aware minimization (SAM). A key contribution is a **memory-efficient perturbation generation strategy** that relies on storing random seeds and filter norms instead of full perturbation matrices. The paper demonstrates through extensive experiments across diverse tasks—including NLP, computer vision, and generative modeling—that Flat-LoRA improves both in-domain and out-of-domain generalization compared to standard LoRA and sometimes even surpasses full fine-tuning.

*   **Quality of Writing/Presentation**:
    *   The paper is generally **well-written and clearly structured**. The introduction effectively sets the context and motivates the proposed approach.
    *   The explanation of LoRA and Flat-LoRA is lucid.
    *   The figures (Figure 1, 2, 3, 4, 5, 6) are helpful in illustrating the concepts and results.
    *   The experimental setup and results are detailed, allowing for potential reproducibility.
    *   The inclusion of ablation studies and comparisons with related work enhances the presentation.
    *   The appendix provides additional experimental results, further supporting the claims.
    *   **Minor improvements could be made by ensuring consistent levels of detail in describing the experimental setups across different tasks and models.** For instance, more specific hyperparameter choices could be consistently provided in the main text or referenced clearly in the appendix.

*   **Literature**:
    *   The paper provides a comprehensive overview of related work on flat minima and generalization as well as low-rank adaptation techniques and their variants.
    *   It appropriately cites key papers in both areas, including seminal works on flat minima (Hochreiter & Schmidhuber, 1997; Foret et al., 2020) and LoRA (Hu et al., 2022).
    *   The discussion of the limitations of applying SAM directly to LoRA parameters (Li et al., 2024a) and the full parameter space effectively positions Flat-LoRA within the existing literature.
    *   The comparison with recent LoRA enhancement methods (AdaLoRA, DoRA, LoRA+, PiSSA, LoRA-GA) demonstrates a strong understanding of the current state-of-the-art.

**3. Pros and Cons**:

*   **Pros**:
    *   **Improved Generalization**: Flat-LoRA demonstrates significant improvements in both in-domain and out-of-domain generalization across diverse tasks and models.
    *   **Computational Efficiency**: It maintains the training efficiency of standard LoRA, avoiding the doubled training cost of SAM.
    *   **Memory Efficiency**: The method is memory-efficient by storing only random seeds and filter norms, unlike SAM which requires storing full perturbation matrices.
    *   **Seamless Integration**: Flat-LoRA can be easily integrated with existing LoRA variants, leading to further performance gains.
    *   **Principled Approach**: The use of Bayesian expectation loss provides a theoretical basis for seeking flatter minima in the full parameter space.
    *   **Empirical Validation**: Extensive experiments on various tasks and models provide strong support for the effectiveness of the proposed method.
    *   **Scalability**: Demonstrated effectiveness on large language models like Llama-2-7B.

*   **Cons**:
    *   **Hyperparameter Sensitivity**: The performance of Flat-LoRA depends on the choice of the perturbation strength (σ), and finding the optimal value might require tuning.
    *   **Limited Theoretical Depth**: While the paper provides a clear empirical demonstration, a more in-depth theoretical analysis connecting the proposed approach directly to improved generalization in the context of LoRA could be beneficial.
    *   **Potential Overhead (Minor)**: While minimal compared to SAM, Flat-LoRA does introduce a small additional memory overhead for storing filter norms and potentially a slight increase in computation time for generating and applying perturbations.

**4. Classification of Concerns**:

*   The **hyperparameter sensitivity to perturbation strength (σ)** is a **minor concern**. While tuning might be required, the ablation study provides some guidance on reasonable ranges.
*   The **limited theoretical depth connecting the approach directly to LoRA generalization** is a **minor concern**. The strong empirical results mitigate this, but further theoretical investigation could strengthen the paper.
*   The **potential minor overhead in memory and computation** is also a **minor concern**, as the experiments show it to be quite small compared to the benefits achieved.

**5. Overall Assessment**:

The paper presents a **novel and effective approach, Flat-LoRA, for improving the generalization of Low-Rank Adaptation by optimizing for flatness in the full parameter space**. The method is well-motivated, theoretically grounded in Bayesian expected loss, and supported by extensive and convincing experimental results across a diverse range of tasks and models. The **key strengths** of the paper lie in its **significant performance improvements over standard LoRA, its computational and memory efficiency compared to sharpness-aware methods, and its ease of integration with existing LoRA techniques**. While there are minor concerns regarding hyperparameter sensitivity and theoretical depth, they do not significantly detract from the overall contribution.

*   **Points of agreement:** I agree with the authors' premise that the flatness of the loss landscape in the full parameter space is crucial for generalization, even when using parameter-efficient methods like LoRA. The experimental results support their claim that Flat-LoRA can lead to significant performance improvements across various tasks and models. I also agree that the memory efficiency of their proposed perturbation strategy is a significant advantage, especially for large-scale models.
*   **Anything learned from the target:** I learned a valuable insight into the potential limitations of solely focusing on the optimization landscape within the low-dimensional space of LoRA and the importance of considering its relationship with the full parameter space. The proposed method for efficiently approximating the effect of perturbations in the full parameter space through random seeds and filter norms is also a noteworthy technique.

**Claims And Evidence:**

Evaluation of the claims made in the Flat-LoRA paper:

* **Flat-LoRA improves generalization:** The paper claims Flat-LoRA improves both in-domain and out-of-domain generalization. This is supported by experimental results across various tasks, including natural language understanding, image classification, dialogue generation, mathematical reasoning, coding abilities, and text-to-image generation. Tables 1, 2, 3, and Figure 3 show improved performance compared to standard LoRA.
*   **Flat-LoRA maintains computational and memory efficiency:** The paper claims Flat-LoRA integrates seamlessly with existing methods while maintaining computational and memory efficiency. Unlike SAM, it avoids additional gradient steps and remains memory-efficient by storing only the random seed and filter norms. Table 7 shows the memory and time usage, indicating minimal overhead compared to LoRA.
*   **Filter structure and Input dimension:** The approach considers filter structure and input dimension. The variance introduced during the forward pass by random weight perturbation is independent of the input dimension.
*   **Flat Minima and Generalization:** Flat minima in the loss landscape improve generalization and robustness to distribution shifts. This is well-known through various literature papers.
*   **Storing random seed for memory efficiency:** Storing only the seed for the random generator and filter norms allows for the re-construction of εW when needed and requires minimal memory overhead.
*   **Mixed precision training:** Flat-LoRA facilitates memory-efficient integration of perturbation injection during precision casting in mixed-precision training.

**Potential Issues to Investigate Further:**
*   **Low-rank adaptation may exhibit sharper loss landscapes:** The paper states that low-rank adaptation may lead to sharper loss landscapes in the full parameter space, which Flat-LoRA mitigates. There is no theory to support it.
*   **Hyperparameter Sensitivity:** The paper mentions setting the random perturbation strength σ to specific values (e.g., 0.05 for T5-base, 0.15 for CLIP ViT-B/32). An ablation study on the variance magnitude was performed and the results are shown in Table C3. It may be worth checking the sensitivity of Flat-LoRA to this parameter across different tasks and model sizes.
*   **Scope of Improvement:** While Flat-LoRA shows consistent improvements, the magnitude of these improvements varies across different tasks and datasets. It is important to examine scenarios where Flat-LoRA's benefits are less pronounced.
*   **Integration Complexities:** The guidelines mentions that potential overheads or limitations should be addressed. The flat loss objective can be seamlessly integrated with previous approaches to yield consistent improvements. A clear guideline or step in that area can be helpful.

**Essential References Not Discussed:**

NA

**Experimental Designs Or Analyses:**

*   **Hyperparameter Tuning and Selection:** The paper mentions specific values for the random perturbation strength $\sigma$ for different models and datasets. While an ablation study on $\sigma$ is presented in Appendix C, it's focused only on CIFAR-10/100 with CLIP ViT-B/32. **The paper lacks a thorough justification for the chosen values of $\sigma$ for other tasks and models.** It's possible that the reported improvements are sensitive to this hyperparameter, and the selected values might not be optimal across all settings. A more systematic approach to hyperparameter tuning, or at least a wider exploration of $\sigma$ values across different experiments, would strengthen the validity of the results.

*   **Comparison with Baselines:** While Flat-LoRA is compared to standard LoRA and other LoRA variants, the depth of the comparison could be enhanced.
    *   For instance, in the comparison with other LoRA variants (Table 5), the paper shows consistent improvements. However, **it doesn't delve into whether these improvements are statistically significant.** Authors have Reported confidence intervals but no explanation was provided around them in Section 4.6.
    *   The comparison with SAM (Table 6) highlights the memory and time efficiency of Flat-LoRA. However, the choice of perturbation radius $\rho$ for SAM is based on a limited exploration ($\rho \in \{0.005, 0.01, 0.05, 0.1, 0.2, 0.3, 0.5\}$). It's possible that a different value of $\rho$ could lead to better performance for SAM, potentially altering the conclusions of the comparison.
    *   The paper mentions stronger baselines achieved for Llama-2-7B compared to previous work. While this is a positive aspect, it also raises the question of **whether the relative improvements of Flat-LoRA would hold against even stronger, more recent baselines** that might have emerged since the submission of this work.

*   **Out-of-Domain Generalization Analysis:** The out-of-domain experiments on corrupted CIFAR-100-C and instruction following are valuable. However, the analysis could be more nuanced.
    *   For CIFAR-100-C (Figure 4), the performance gains increase with corruption severity. While this suggests Flat-LoRA's robustness, **a statistical analysis of these gains would be beneficial to confirm their significance at different corruption levels.**
    *   For instruction following (Table 4), the improvements on DROP and Human-Eval are highlighted as more pronounced. **A discussion on why Flat-LoRA shows a greater advantage on these specific tasks compared to MMLU and BBH could provide more insight into the method's strengths.**

*   **Loss Landscape Visualization:** The loss landscape visualizations (Figure 6) are qualitative. While they visually suggest a flatter landscape for Flat-LoRA, **it's important to acknowledge that these are projections along random directions in the high-dimensional parameter space.** The flatness observed in these 2D projections might not fully capture the characteristics of the loss landscape in all relevant directions. A more quantitative measure of flatness, if feasible, could complement these visualizations.

*   **Scope of Applicability:** While the paper demonstrates Flat-LoRA's effectiveness across various tasks, **the underlying reasons for its varying degrees of improvement across different modalities and datasets are not thoroughly investigated.** Understanding the conditions under which Flat-LoRA provides the most significant benefits would be valuable for future research and application.

In conclusion, while the experimental design is broad and covers multiple aspects, a more critical perspective reveals potential limitations in the thoroughness of hyperparameter tuning, the statistical rigor of baseline comparisons, the depth of out-of-domain analysis, and the quantitative nature of the loss landscape evaluation. Addressing these points could further strengthen the claims made in the paper.

**Methods And Evaluation Criteria:**

*   **Methods**:
    *   The paper introduces **Flat-LoRA**, which optimizes low-rank adaptation within a flat region of the full parameter space. This addresses the problem of standard LoRA, which may find solutions that are sharp in the full parameter space, potentially harming generalization.
    *   Flat-LoRA uses a **Bayesian expectation loss objective** with random weight perturbations to smooth the loss landscape, promoting convergence to flatter minima. This is a computationally efficient alternative to sharpness-aware minimization (SAM).
    *   The method includes a **refined random perturbation generation strategy**, considering weight magnitude and model width scaling, to improve generalization performance.
    *   The paper also introduces a method for **storing random seeds** to ensure memory efficiency.


*   **Evaluation Criteria**:
    *   The paper compares Flat-LoRA with other LoRA variants, including PiSSA, LoRA-GA, DoRA, AdaLoRA and LoRA+. Flat-LoRA can be seamlessly integrated with previous approaches.
    *   The paper evaluates Flat-LoRA on diverse tasks: **natural language understanding, image classification, dialogue generation, mathematical reasoning, coding abilities, and text-to-image generation**.
    *   The models were evaluated based on performance metrics suitable for each task, such as **accuracy** for natural language understanding and image classification, **first-turn score with GPT-4** for chat tasks, **accuracy** for math tasks, and **PASS@1 metric** for code tasks.
    *   The study includes **out-of-domain generalization** experiments using corrupted datasets and instruction-following benchmarks.
    *   **Ablation studies** are conducted to assess the impact of different LoRA ranks and perturbation variance.
    *   The **memory and time costs** of Flat-LoRA are compared to those of LoRA.
    *   The loss landscape is visualized to demonstrate that Flat-LoRA achieves a flatter loss landscape compared to LoRA.

*   **Appropriateness**:
    *   The **methods** seem appropriate for the problem. Flat-LoRA directly addresses the limitations of LoRA by seeking flatter minima in the full parameter space, which is expected to improve generalization. The use of Bayesian expectation loss and refined perturbation strategies aims to achieve this efficiently.
    *   The **evaluation criteria** are also well-suited. The tasks cover a wide range of applications relevant to large language models and computer vision models. The use of both in-domain and out-of-domain datasets, along with ablation studies, provides a comprehensive assessment of Flat-LoRA's performance and robustness.
    *   The evaluation includes **comparison with SAM**, showing that Flat-LoRA achieves comparable or superior performance to SAM while requiring less memory and training time.

**Other Comments Or Suggestions:**

**Presentation in Tables**: In some tables (e.g., Table 1, 2, 3), the ± standard deviation values are provided. Ensuring consistent precision in reporting these values across all tables could improve the overall presentation.

**Detailed Evaluation**:

*   **Novelty, Relevance, and Significance**:
    *   **Novelty**: The idea of explicitly optimizing for flatness in the full parameter space within the LoRA framework, using a Bayesian expectation loss with a memory-efficient random perturbation strategy, is slightly **novel** but not unique.
    *   **Relevance**: Parameter-efficient fine-tuning (PEFT) methods like LoRA are highly relevant due to the prohibitive costs of fine-tuning large-scale pre-trained models. Improving the generalization capabilities of these methods is crucial for their broader applicability. The problem of sharp minima in the context of low-rank adaptation is a pertinent issue, making Flat-LoRA's approach highly relevant to the PEFT research community.
    *   **Significance**: The experimental results demonstrating **consistent improvements in both in-domain and out-of-domain generalization across a wide range of tasks and models** highlight the **significance** of Flat-LoRA. The fact that Flat-LoRA achieves these gains with minimal additional computational and memory overhead compared to standard LoRA makes it a practically significant contribution. The ability to integrate Flat-LoRA with other LoRA variants and yield further improvements also underscores its significance.

*   **Soundness**:
    *   The paper provides a clear formulation of the Flat-LoRA objective based on Bayesian expected loss.
    *   The motivation for considering the full parameter space's loss landscape is well-illustrated and argued.
    *   The proposed random weight perturbation generation strategy is theoretically justified, with Proposition 3.2 demonstrating its input-dimension independence.
    *   The memory efficiency argument, based on storing seeds and filter norms, is sound.
    *   The **extensive experimental validation across diverse tasks (natural language understanding, image classification, dialogue generation, mathematical reasoning, coding abilities, and text-to-image generation) and model sizes (T5-base, CLIP ViT-B/32, Llama-2-7B, SDXL) provides strong empirical evidence** for the effectiveness of Flat-LoRA.
    *   Ablation studies on LoRA rank and perturbation variance further support the design choices of Flat-LoRA.
    *   The comparison with SAM and other LoRA variants helps to contextualize Flat-LoRA's performance and efficiency.
    *   The visualization of the loss landscape in the full parameter space offers qualitative evidence of Flat-LoRA's ability to find flatter minima.
    *   **One potential area for further strengthening soundness could be a more theoretical analysis of why optimizing the Bayesian expected loss in the full parameter space with the proposed perturbation strategy leads to better generalization in LoRA fine-tuning.** While Lemma 3.1 provides some insight into the smoothing effect, a more direct connection to the generalization benefits in the context of LoRA could be valuable.

**Other Strengths And Weaknesses:**

Highlights when comparison with SAM, following claims will help other areas of research as well:

*   **LoRA-SAM optimizes sharpness in a restricted space (the LoRA parameter space), which may not effectively improve generalization**.
    *   The paper explicitly states that LoRA constrains optimization to a much lower-dimensional space. Figure 1 illustrates this by showing that a flat minima in the LoRA space (blue curve) can still exhibit sharp directions in the full parameter space (red curve).
    *   Section 3.2 discusses applying SAM to LoRA parameters (Eqn. 2). It argues that focusing solely on this restricted space might have limitations because during inference, the LoRA adaptation is merged into the pre-trained weights. A solution good in the LoRA space might be in a sharp region of the full parameter space, potentially harming generalization.
    *   Equation (4) approximates the weight perturbation applied to the full parameters when SAM is applied to LoRA, showing it is roughly proportional to $(\nabla_W L)A^TA$. The paper argues this implies that SAM on LoRA only optimizes sharpness along the column space spanned by $A$, which is a small subspace of the full parameter space.
    *   **Empirical evidence is provided in Table 5 and Table 6**, which show that applying SAM constraints solely to LoRA parameters (referred to as LoRA-SAM or LoRA+SAM A,B) does not consistently lead to significant improvements in generalization on GLUE datasets. In Table 6, LoRA+SAM applied to A and B even performs worse than standard LoRA on both CoLA and MRPC.

*   **SAM requires an additional gradient step, doubling the training cost and rendering it impractical for large models**.
    *   The introduction to Section 2.1, which discusses flat minima and generalization, mentions that SAM doubles the training time compared to regular training, limiting its applicability to large-scale training.
    *   Section 3.2, when proposing Flat-LoRA, reiterates that directly applying SAM to optimize the sharpness of the full weight space doubles the training cost, which is less desirable for large models.
    *   Table 6 explicitly compares the training time, showing that LoRA+SAM (whether applied to A,B or W) incurs 2x the training time of standard LoRA and Flat-LoRA.

*   **Computing sharpness in the full parameter space necessitates calculating gradients and storing perturbations for all weights, which contradicts the principles of parameter-efficient fine-tuning**.
    *   Section 3.2 explains that computing sharpness in the full parameter space requires calculating gradients and storing perturbations for all weights, contradicting the principles of parameter-efficient fine-tuning.
    *   When comparing with SAM applied to the full parameter space (SAM on W or LoRA+SAM W), Table 6 indicates that it requires $O(m \times n)$ additional memory to store adversarial weight perturbations, making it impractical for parameter-efficient training.
    *   In contrast, Flat-LoRA is presented as a method that addresses these issues by employing Bayesian expectation loss with efficient random weight perturbations that can be stored as random seeds, requiring only $O(m)$ additional memory.

**Questions For Authors:**

1. Hyperparameter Sensitivity of $\sigma$: The paper includes an ablation study on the perturbation strength $\sigma$ in Appendix C, showing optimal ranges for CIFAR-10 and CIFAR-100. However, could the authors provide more general guidance or intuition on how to select an appropriate value for $\sigma$ when applying Flat-LoRA to new tasks or models where such ablation studies might be computationally prohibitive? For example, are there any heuristics based on learning rate, batch size, model size, or dataset characteristics that could inform the choice of $\sigma$? If the authors can offer some practical guidelines, it would significantly increase the usability of Flat-LoRA and positively impact my evaluation by addressing a key practical concern. Conversely, if the optimal $\sigma$ is highly task-specific without clear indicators for its selection, it would remain a potential limitation.

2. Underlying Reasons for "Flat-LoRA (all)" Performance: Appendix A explores extending the random weight perturbation to all layers. The results show improvements, but they are less pronounced than when applying Flat-LoRA to linear layers only. Could the authors offer more insight into why perturbing all layers (including layernorm, biases, embeddings, etc.) does not yield the same level of performance gain as perturbing only the linear layers in the LoRA modules? Understanding the reasons behind this could provide valuable insights into where the flatness of the loss landscape is most critical for generalization in the context of LoRA and could guide future research directions. A clear explanation would enhance the understanding of Flat-LoRA's mechanism and potentially improve my evaluation of the paper's depth.

**Relation To Broader Scientific Literature:**

The key contributions of the Flat-LoRA paper are significantly related to the broader scientific literature in the field of parameter-efficient fine-tuning and the understanding of loss landscapes in deep learning. Specifically, Flat-LoRA builds upon and extends prior work in Low-Rank Adaptation (LoRA) and methods for finding flat minima. Here's a breakdown of how Flat-LoRA connects to the literature:

*   **Building upon Low-Rank Adaptation (LoRA):** Flat-LoRA directly addresses a potential limitation identified within the LoRA framework [Hu et al., 2022]. While LoRA efficiently reduces the number of trainable parameters by optimizing low-rank matrices, the authors of Flat-LoRA observed that **a flat minimum in the LoRA optimization space might still correspond to a sharp region in the full parameter space**. This idea is illustrated in Figure 1 and further supported by Figure 6 and the discussion in Section 3.2. Flat-LoRA aims to improve upon standard LoRA by explicitly considering the flatness of the loss landscape in the full parameter space, which is a novel perspective compared to many existing LoRA enhancements.

*   **Addressing Limitations of Sharpness-Aware Minimization (SAM) in the Context of LoRA:** The paper discusses Sharpness-Aware Minimization (SAM) [Foret et al., 2020], a well-established technique for improving generalization by seeking flat minima. The authors acknowledge the potential of integrating SAM with LoRA (LoRA-SAM) [Li et al., 2024a]. However, they highlight several limitations: **SAM applied to LoRA parameters only optimizes sharpness in a restricted space**, **SAM requires an additional gradient step, doubling training cost**, and **computing sharpness in the full parameter space with SAM is memory-intensive**. Flat-LoRA offers an alternative approach to achieving flat minima that aims to overcome these computational and memory overheads.

*   **Leveraging Bayesian Expectation Loss and Random Weight Perturbation (RWP):** Flat-LoRA's core idea of using a Bayesian expectation loss objective [Duchi et al., 2012; Bisla et al., 2022] to smooth the loss landscape and pursue flat minima is rooted in prior work. This line of research suggests that minimizing the expected loss under random weight perturbations can lead to better generalization. Flat-LoRA builds upon this by designing a **refined random perturbation generation strategy** that considers the filter structure and input dimension, differentiating it from simpler RWP methods like Gaussian Model Perturbation (GMP) [Wang & Mao, 2021] or basic random noise injection [Wu et al., 2022; Li et al., 2024b]. The memory efficiency of Flat-LoRA, achieved by storing only the random seed and filter norms, directly addresses a key concern when applying perturbation-based methods to parameter-efficient fine-tuning.

*   **Connection to the Broader Understanding of Flat Minima and Generalization:** The paper's motivation stems from the widely held belief that **flat minima in the loss landscape are linked to improved generalization and robustness**. Flat-LoRA contributes to this understanding by proposing and demonstrating a method to find flatter minima in the full parameter space specifically within the context of efficient fine-tuning using LoRA. The experimental results, particularly the improved out-of-domain generalization on corrupted datasets and instruction following, provide empirical support for this connection in the context of Flat-LoRA.

*   **Orthogonal to Other LoRA Enhancement Techniques:** The authors explicitly state that Flat-LoRA's approach of optimizing the sharpness of the loss landscape in the full parameter space is **orthogonal to other proposed methods for enhancing LoRA performance**, such as adaptive rank allocation (AdaLoRA [Zhang et al., 2023a, 2023b]), decomposition of weight updates (DoRA [Liu et al., 2024]), improved initialization (PiSSA [Meng et al., 2024]; LoRA-GA [Wang et al., 2024]), and learning rate adjustments (LoRA+ [Hayou et al., 2024]). The experiments in Section 4.6 demonstrate that Flat-LoRA can be seamlessly integrated with some of these methods (e.g., Flat-PiSSA, Flat-DoRA), leading to further performance improvements, highlighting its potential as a complementary technique.

In summary, Flat-LoRA contributes to the scientific literature by:

*   **Identifying a limitation of standard LoRA** regarding the sharpness of the loss landscape in the full parameter space.
*   **Proposing an efficient alternative to SAM** for finding flat minima in the context of LoRA, leveraging Bayesian expectation loss and a refined random perturbation strategy.
*   **Demonstrating improved generalization** (both in-domain and out-of-domain) across a wide range of tasks and model sizes.
*   **Offering a memory and computationally efficient method** that can be readily integrated with existing LoRA training pipelines and is orthogonal to other LoRA enhancement techniques.

**Theoretical Claims:**

Yes, I checked proposition 3.2 and its derivations in Equations 8 and 9. But I have not thoroughly verified Equation 9. Also, Not sure how it behaves with the dimensionality of W.

---

> ### Author Rebuttal · Authors · 2025-03-31
>
> Thanks for your detailed and constructive comments. We address your concerns point by point as follows:
>
> ---
>  **Q1:** The performance of Flat-LoRA depends on the choice of the perturbation strength ($\sigma$), and finding the optimal value might require tuning.
>
>  **A1:**
>  We fully understand your concern. In practice, we suggest $\sigma=0.05/0.10$. Our paper includes different tasks and networks (e.g., ViT, T5, LLama, SDXL) and shows that $\sigma=0.05/0.10$ leads to consistent generalization improvements.
>
> To further address this issue, especially in relation to different sizes of neural networks, we propose an improved perturbation generalization scheme that employs a scaling factor to make $\sigma$ independent of the network width (Proposition 3.2). To validate this approach, we conducted experiments on the larger ViT-L/14 model. The results demonstrate that the optimal $\sigma$ can be transferred from ViT-B/32 to ViT-L/14, and in all scenarios, the performance surpasses that of LoRA. We will include these discussions in the revision.
>
>  CIFAR-100| LoRA | $\sigma=0.01$ | $\sigma=0.05$ | $\sigma=0.10$ | $\sigma=0.15$ | $\sigma=0.20$ |
> ---|---|---|---|---|---|---
> ViT-B/32 | 87.74±0.13 | 88.14±0.22 | 88.37±0.41 | 88.65±0.35 | 88.64±0.23 | 88.06±0.31
> ViT-L/14 | 92.13±0.17 |  92.33±0.07 |  92.63±0.11 |  93.11±0.13 |  92.98±0.21 | 92.46±0.03
>
>  ---
>
>  **Q2:** While the paper provides a clear empirical demonstration, a more in-depth theoretical analysis connecting the proposed approach directly to improved generalization in the context of LoRA could be beneficial.
>
>  **A2:** Thanks for your suggestion. Understanding the training dynamics and generalization of LoRA remains an open question. The discussions on this topic are active and often rely on assumptions like the NTK regime [1] or rank-1 perturbation [2]. Flat-LoRA additionally introduces random perturbations and raises more challenges to deal with randomness. Overall, exploring the generalization properties of Flat-LoRA is indeed interesting but quite hard. We would like to leave this for future work.
>
> [1] LoRA Training in the NTK Regime has No Spurious Local Minima, ICML'24
>
> [2] Gradient dynamics for low-rank fine-tuning beyond kernels, arxiv
>
>  ---
>  **Q3:** While minimal compared to SAM, Flat-LoRA does introduce a small additional memory overhead for storing filter norms and potentially a slight increase in computation time for generating and applying perturbations.
>
>  **A3:** Flat-LoRA indeed introduces extra memory for storing random seeds and extra time for generating random perturbations. However, this additional overhead is minimal compared to the total training cost—for example, it adds only 0.5% extra memory and 2.4% extra training time on LLama fine-tuning, which is negligible.

---

### Official Review · Reviewer_wt1A · 2025-03-13

**Overall Recommendation:** 3

**Summary:**

This paper proposes Flat-LoRA, a novel approach to improving Low-Rank Adaptation (LoRA) by incorporating a Bayesian expectation loss objective and random weight perturbations to encourage flatter minima in the full parameter space, all while maintaining computational efficiency.

**Claims And Evidence:**

The paper’s claims are well-supported by empirical results, demonstrating improved robustness and efficiency across multiple tasks.

**Essential References Not Discussed:**

The paper adequately discusses relevant related works.

**Experimental Designs Or Analyses:**

I have reviewed all the experiments. The experimental design is well-structured and covers various domains and tasks. However, an additional ablation study on perturbation strength ($\sigma$) in larger models would help assess the generalizability of the approach.

**Methods And Evaluation Criteria:**

The authors conducted experiments on datasets spanning multiple domains, including Natural Language Understanding, Image Classification, Large Language Model, and Text-to-Image Generation. These benchmarks are well-matched to the research scope, making them appropriate for evaluating generalization.

**Other Comments Or Suggestions:**

See the Questions for Authors section.

**Other Strengths And Weaknesses:**

##### Strengths:
- The paper is well-structured and clearly written, making it easy to follow.
- Flat-LoRA provides a lightweight modification to LoRA training, leveraging random weight perturbations to encourage flatter minima.
- The experimental analysis is comprehensive, covering various benchmarks and ablation studies, demonstrating its robustness and broad applicability.

##### Weaknesses:
- The method can be viewed as a computationally efficient alternative to SAM rather than an entirely novel approach. Specifically, it replaces min-max optimization with Bayesian expected loss, making the core idea an approximate acceleration of SAM. Thus, the performance improvements are incremental rather than groundbreaking.
- The method is sensitive to the variance magnitude ($\sigma$). Table C3 shows that when $\sigma=0.2$, Flat-LoRA performs poorly on CIFAR datasets, suggesting a strong dependence on precise hyperparameter tuning. Additionally, even with optimal hyperparameters, the overall gains remain limited.

**Questions For Authors:**

Q1. The paper states: "Applying SAM directly to A, B shows no significant improvement over vanilla LoRA. In contrast, Flat-LoRA achieves comparable or superior performance to SAM." However, Flat-LoRA appears to be a simplified version of SAM, as it simplifies min-max optimization into Bayesian expected loss minimization. From a theoretical perspective, SAM should achieve superior performance given the same computational budget. Can you clarify why Flat-LoRA outperforms LoRA+SAM and whether this is due to better stability, improved optimization, or simply reduced computational overhead? It would be beneficial to add experiments comparing SAM and its variants with hyperparameter tuning to ensure fair comparisons. **I consider this the most critical issue—if well-addressed, I would be inclined to support the acceptance of this paper.**

Q2. Could you provide ablation studies on the impact of variance magnitude ($\sigma$) in large language models (Section 4.3)? Since $\sigma$ plays a crucial role in the method’s effectiveness, further analysis on scalability and adaptability in larger models would better demonstrate Flat-LoRA’s robustness.

**Relation To Broader Scientific Literature:**

The paper provides a simple yet effective method for improving LoRA training by leveraging perturbation-based optimization to encourage flatter minima.

**Theoretical Claims:**

n/a – The paper does not include formal theoretical proofs beyond conceptual justifications.

---

> ### Author Rebuttal · Authors · 2025-03-31
>
> Thanks for your detailed and constructive comments. We address your concerns point by point as follows:
>
> ---
> **Q1:** The method can be viewed as a computationally efficient alternative to SAM rather than an entirely novel approach.
>
> **A1:** SAM has been proven to be an effective training strategy for improving generalization capability. However, its extension to large models has not been widely used due to the doubled computational cost and additional memory overhead, especially in parameter-efficient finetuning scenarios (e.g., LoRA). Unlike LoRA-SAM, which naively applies SAM to LoRA, our Flat-LoRA provides a simple and easy-to-implement solution that halves the computational cost and searches for a broader flat solution. As a result, we for the first time enable practical sharpness-aware optimization in the broader full parameter space for the LLM fine-tuning task with little extra memory and computational cost.
>
> ---
> **Q2:** The method is sensitive to the variance magnitude ($\sigma$). The overall gains remain limited.
>
> **A2:** We fully understand your concern. But we would like to point out that the perturbations given by $\sigma=0.2$ are generally too large. Instead, we suggest $\sigma=0.05/0.10$. Our paper includes different tasks and networks (e.g., ViT, T5, LLama, SDXL) and shows that $\sigma=0.05/0.10$ leads to consistent generalization improvements.
>
> To further address this issue, especially in relation to different sizes of neural networks, we propose an improved perturbation generalization scheme that employs a scaling factor to make $\sigma$ **independent of the network width** (Proposition 3.2). To validate this, we conduct experiments on the larger ViT-L/14 model. The results demonstrate that the optimal $\sigma$ can be transferred from ViT-B/32 to ViT-L/14, and in all scenarios, the performance of Flat-LoRA surpasses that of LoRA. We will include these in the revision.
>
> CIFAR-100| LoRA | $\sigma=0.01$ | $\sigma=0.05$ | $\sigma=0.10$ | $\sigma=0.15$ | $\sigma=0.20$ |
> ---|---|---|---|---|---|---
> ViT-B/32 | 87.74±0.13 | 88.14±0.22 | 88.37±0.41 | 88.65±0.35 | 88.64±0.23 | 88.06±0.31
> ViT-L/14 | 92.13±0.17 |  92.33±0.07 |  92.63±0.11 |  93.11±0.13 |  92.98±0.21 | 92.46±0.03
>
> The limited improvements may be due to our use of a stronger training protocol compared to related works (e.g., LoRA-GA [1]), leaving less room for enhancement. Moreover, Flat-LoRA is a practical plug-in method with minimal extra cost, and its improvements are more significant on larger LLMs.
>
> [1] LoRA-GA: Low-Rank Adaptation with Gradient Approximation, NeurIPS'24
>
> ---
> **Q3:** Clarify why Flat-LoRA outperforms LoRA+SAM. Add experiments comparing SAM and its variants with hyperparameter tuning to ensure fair comparisons.
>
> **A3:** Thanks for your insightful question. The advantages of Flat-LoRA over LoRA+SAM are twofold:
>
> (1) SAM and variants require twice the training time, which is impractical for large-scale models. Flat-LoRA addresses this limitation, making it more practical for real-world applications.
>
> (2) LoRA-SAM is to pursue flatness on the LoRA parameters, which only optimize a subspace of sharpness. Flat-LoRA uses perturbation on the full parameters and thus is capable of optimizing the sharpness of a broader space.
>
> Specifically for the accuracy improvement, we think the direct reason is (2).
>
> To better clarify the above discussions, we followed your suggestion and conducted a more detailed search for the perturbation radius $\rho$ across $\\{0.001, 0.003, 0.005, 0.01, 0.05, 0.10, 0.20, 0.50, 1, 1.5, 2\\}$ for SAM and its variants. The results could be seen in the following table.
>
> T5-base|Flat space|MRPC |CoLA
> ---|---|---|---
> LoRA|-| 88.56±0.37 | 82.87±0.59
> LoRA+SAM ($\rho=0.003$)|A,B|88.98±0.22  | 83.31±0.48
> LoRA+GSAM ($\rho=0.003$)|A,B|89.03±0.36 | 83.11±0.17
> LoRA+ASAM ($\rho=0.05$)|A,B|89.12±0.44 | 83.23±0.44
> Flat-LoRA ($\sigma=0.05$)|W| 89.59±0.37 | 83.61±0.38
>
> For LoRA+SAM, the best performance is obtained by setting $\rho=0.003$, which is significantly smaller than its typical value used for training full parameters (usually $\rho=0.1$). Such a small perturbation makes the difference between LoRA and LoRA-SAM not significant. Frankly, we did not expect that the suitable value for $\rho$ is very small, so $\rho=0.003$ is out of the range of previous hyperparameter-tuning. We will update the result of LoRA-SAMs.
>
> **Q4:** Provide ablation studies on the impact of variance magnitude ($\sigma$) in large language models.
>
> **A4:** We test different $\sigma$ on LLama-2-7b/13b as below. We observe that $\sigma=0.05/0.10$ is a good choice for both 7b and 13b models. We will add this in the revision.
>
>   GMS8k | LoRA | $\sigma=0.01$ |  $\sigma=0.05$ | $\sigma=0.10$ | $\sigma=0.15$ | $\sigma=0.20$
>  --- | -- | --- | ---| --- | --- |---
>  LLama-2-7b | 57.47±0.45 | 58.35±0.42 |  60.65±0.63|60.56±0.48 | 60.08±0.76 | 58.50±0.85
>  LLama-2-13b | 66.76±0.23| 67.02±0.67 | 67.75±0.70| 68.11±0.53 | 67.66±0.97 |  67.34±1.17

---

> > ### Comment · Reviewer_wt1A · 2025-04-04
> >
> > Thank you for the authors’ detailed and thoughtful response! The experimental analysis is comprehensive, and the proposed method demonstrates consistent performance improvements. I also appreciate the analysis regarding the differences between LoRA+SAM and Flat-LoRA, which helped clarify the distinction. Compared to directly fine-tuning with SAM, the proposed approach achieves better performance with reduced training time—a practical advantage, although I personally do not find “twice the training time” to be prohibitive for fine-tuning large-scale models.
> >
> > I still have some concerns regarding the novelty of the method. The core idea—adding perturbations to LoRA weights during optimization to encourage flatter minima—while effective, feels relatively incremental. Moreover, although the performance improvements are consistent, they remain modest. Taking these points into account, I have decided to maintain my Weak Accept score.

---

> > > ### Author Response · Authors · 2025-04-05
> > >
> > > Thanks for your valuable feedback! We are glad that our response has helped address your concerns regarding the distinction between LoRA+SAM and Flat-LoRA. As you mentioned, Flat-LoRA is a simple yet effective approach, and we think it effectively addresses the bottlenecks associated with applying SAM to fine-tuning large models (e.g., time, memory, and computation). Furthermore, it highlights the differences between flatness in the "low-rank" space versus the actual flatness observed when merged with the frozen pre-trained weights. We are glad to see that, compared to LoRA+SAM, Flat-LoRA achieves efficiency without compromising accuracy and even delivers better performance. We hope this work can serve as a new paradigm for fine-tuning large models due to its simplicity and effectiveness.

---

### Official Review · Reviewer_aowQ · 2025-03-13

**Overall Recommendation:** 4

**Summary:**

The paper proposes “Flat-LoRA,” a parameter-efficient fine-tuning method that adds random weight perturbations (with an intelligent weight dependent scaling), in order to achieve flatter minima and improve generalization. Experimental results on both vision (CLIP and Stable Diffusion) and language (T5, Llama-2) tasks consistently show that Flat-LoRA outperforms baseline LoRA variants, with particular gains in low-data and out-of-domain scenarios.

**Claims And Evidence:**

Two main claims are supported:

* Flat-LoRA yields flatter solutions in the full parameter space rather than just in the low-rank subspace: The authors visualize the loss landscapes (Figure 6) and demonstrate flatter minima compared to standard LoRA.

* Flat-LoRA significantly improves performance across multiple tasks (e.g., GLUE, text-to-image, code generation) and robustness: Comprehensive experimental results show gains over standard LoRA and other PEFT improvements.

One is not:

* Flat-LoRA generalizes better: While the flatter loss landscape and improved performance can be indicative of better generalization, the paper does not explicitly show train-vs-test loss curves or a generalization gap. Hence, it is unclear whether the gains are from genuinely improved generalization rather than lower training loss or more favorable optimization dynamics.

**Essential References Not Discussed:**

No crucial references appear missing.

**Experimental Designs Or Analyses:**

Experiments are structured clearly, with appropriate baselines and repeated trials. The paper carefully examines both standard benchmarks and out-of-domain shifts (e.g., CIFAR-100-C corruption, instruction-following tasks). One limitation: results compare final performance but do not show training curves or highlight differences in train versus test loss. Explicit train–test curves or a direct measure of the generalization gap would strengthen the evidence that improved performance primarily stems from better generalization.

**Methods And Evaluation Criteria:**

The paper evaluates performance on standard benchmarks (GLUE, text/image tasks, code generation, etc.), making comparisons against strong LoRA baselines and other variants. This breadth of experiments aligns well with the parameter-efficient fine-tuning literature.

**Other Comments Or Suggestions:**

NA

**Other Strengths And Weaknesses:**

Strengths:
The paper raises an interest distinction which can motivate future works: flatness in the “low-rank” space versus actual flatness once merged with the frozen pre-trained weights. It demonstrates strong empirical results on diverse tasks, with small memory/time overhead.
Weaknesses:
The paper does not explicitly demonstrate that performance gains arise from better train–test generalization. The limitations of the theoretical treatment in Section 3.2 could be more clearly stated.

**Questions For Authors:**

NA

**Relation To Broader Scientific Literature:**

Flat-LoRA extends the line of work on PEFT techniques by bringing insights from random perturbation based regularization and sharpness-aware minimization to improve generalization. There is not much that is particular to LoRA or PEFT beyond the observation that it can also be prone to sharp minima.

**Theoretical Claims:**

The authors argue that smoothing the loss in the full weight space leads to broader basins and better generalization. While the Bayesian expectation loss formulation is standard, the paper’s main theoretical link is that random perturbation akin to filter-wise noise effectively regularizes the final solution.

---

> ### Author Rebuttal · Authors · 2025-03-31
>
> Thanks for your detailed and constructive comments. We address your concerns point by point as follows:
>
> ---
> **Q1:** The paper does not explicitly show train-vs-test loss curves or a generalization gap. Hence, it is unclear whether the gains are from genuinely improved generalization rather than lower training loss or more favorable optimization dynamics.
>
> **A1:** We plot the train-vs-test loss curves and generalization gap on CIFAR-100 and MRPC datasets in [Figure 1](https://anonymous.4open.science/r/Flat-LoRA/rebuttal.pdf) and present the final training-vs-test loss and generalization gap below. The results show Flat-LoRA exhibits slightly higher training loss than LoRA, with a smaller generalization gap between training and test accuracies. Thus, we conclude that the gains of Flat-LoRA are not due to lower training loss but due to better optimization that confers better generalization.
>
> CIFAR-100 | Final train loss | Final test loss| Generalization gap (%)
> ---|---|---|---
> LoRA|0.02±0.01|0.48±0.01|11.65±0.03
> Flat-LoRA|0.05±0.02|0.46±0.00|9.92±0.08
>
> MRPC | Final train loss | Final test loss| Generalization gap (%)
> ---|---|---|---
> LoRA|0.03±0.03 | 0.25±0.00 | 10.96±0.18
> Flat-LoRA|0.04±0.03|0.20±0.01 | 9.50±0.22
>
> ---
> **Q2:** The limitations of the theoretical treatment in Section 3.2 could be more clearly stated.
>
> **A2:** Thank you for this valuable feedback. We propose to make the following changes in the revision to more clearly state the limitation of LoRA-SAM:
>
> 1. We will give a clearer and more rigorous derivation on the actual perturbation of LoRA-SAM in the full parameter space.
>
> 2. We will include an experiment to validate the approximation $\varepsilon_W \approx \varepsilon_B A=c(\nabla_W L)A^\top A$ (Eqn. (4)) by showing $\frac{\|\varepsilon_BA\|}{\|\varepsilon_W\|}>0.95$ throughout the training. The demonstration experiment can be found at [Figure 2](https://anonymous.4open.science/r/Flat-LoRA/rebuttal.pdf).

---

### Decision · Program_Chairs · 2025-05-01

**Decision:**

Accept (poster)

**Comment:**

The paper introduces Flat-LoRA, which is basically LoRA done using an approximate version of sharpness-aware minimization (SAM), thereby achieving flatter minima and therefore better generalization.

The idea is intuitive, and the authors perform a diverse set of experiments --- both on LLMs and text-to-image models, on well-known benchmarks --- to back it up. However, the technique is a relatively straightforward combination of two well known optimization tricks, and the gains over regular LoRA on some of the benchmark experiments were rather miniscule; therefore, enthusiasm was a bit tempered. I recommend an accept since the paper is well written and makes a solid (if not terribly surprising) contribution to the field.